# Spartan: Differentiable Sparsity
# via Regularized Transportation

**Kai Sheng Tai**
Meta AI

**Taipeng Tian**
Meta AI

**Ser-Nam Lim**
Meta AI

## Abstract

We present Spartan, a method for training sparse neural network models with a predetermined level of sparsity. Spartan is based on a combination of two techniques: (1) soft top-$k$ masking of low-magnitude parameters via a regularized optimal transportation problem and (2) dual averaging-based parameter updates with hard sparsification in the forward pass. This scheme realizes an exploration-exploitation tradeoff: early in training, the learner is able to explore various sparsity patterns, and as the soft top-$k$ approximation is gradually sharpened over the course of training, the balance shifts towards parameter optimization with respect to a fixed sparsity mask. Spartan is sufficiently flexible to accommodate a variety of sparsity allocation policies, including both unstructured and block structured sparsity, as well as general cost-sensitive sparsity allocation mediated by linear models of per-parameter costs. On ImageNet-1K classification, Spartan yields 95% sparse ResNet-50 models and 90% block sparse ViT-B/16 models while incurring absolute top-1 accuracy losses of less than 1% compared to fully dense training.

## 1 Introduction

Sparse learning algorithms search for model parameters that minimize training loss while retaining only a small fraction of non-zero entries. Parameter sparsity yields benefits along several axes: reduced model storage costs, greater computational and energy efficiency during training and inference, and potentially improved model generalization [33]. However, sparse training is a challenging optimization problem: for the general problem of learning parameters $\theta \in \mathbb{R}^d$ subject to the constraint that $\theta$ is $k$-sparse, the feasible set is the union of $\binom{d}{k}$ axis-aligned hyperplanes intersecting at the origin, each of dimension $k$. This complex, nonconvex geometry of the feasible set compounds the existing difficulty of optimizing deep neural network models.

This work focuses on improving the generalization error of sparse neural network models. To this end, we introduce *Spartan*, or *Sparsity via Regularized Transportation*—a sparse learning algorithm that leverages an optimal transportation-based top-$k$ masking operation to induce parameter sparsity during training. Spartan belongs to the family of "parameter dense" training algorithms that maintains a dense parameter vector $\theta \in \mathbb{R}^d$ throughout training [48; 24], in contrast to "parameter sparse" training algorithms that adhere to a $\tilde{\mathcal{O}}(k)$ memory budget for representing the parameters of a $k$-sparse model [3; 31; 32; 13; 16]. While computational cost and memory usage at training time are important considerations, Spartan primarily optimizes for performance at inference time.

Intuitively, Spartan aims to achieve a controlled transition between the *exploration* and *exploitation* of various sparsity patterns during training. In the exploration regime, the goal is for the learner to easily transition between differing sparsity patterns in order to escape those that correspond to poor minima of the loss function. On the other hand, the goal of exploitation is for the learner to optimize model parameters given a fixed sparsity pattern, or a small set of similar patterns. This latter setting

---

Correspondence to: Kai Sheng Tai (`kst@meta.com`)

36th Conference on Neural Information Processing Systems (NeurIPS 2022).

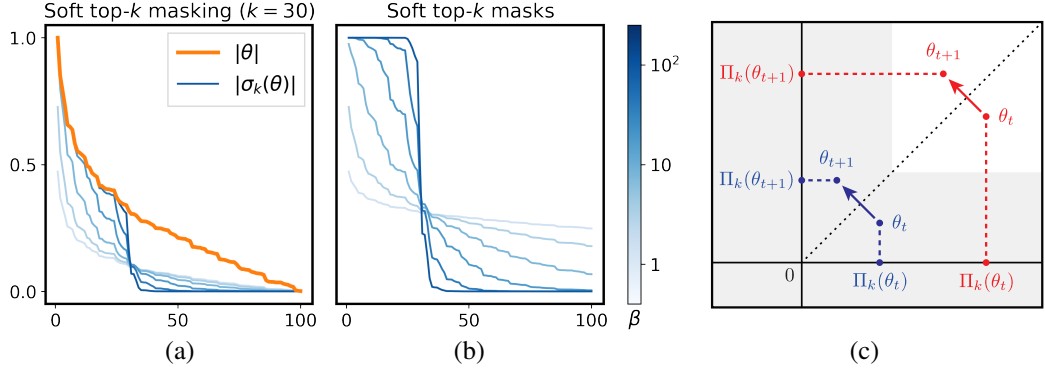

Figure 1: **Soft top-$k$ masking.** (a, b) The soft top-$k$ masking operator $\sigma_k^\beta$ computes approximately $k$-sparse outputs, with the sharpness of the mask controlled by the parameter $\beta$. (c) Small updates to iterates far from the $k$-sparse feasible set ($\theta_t \to \theta_{t+1}$) can correspond to a large perturbation in parameter space after projection by $\Pi_k$. Updates to iterates $\theta_t$ in the approximately sparse region (shaded in grey) correspond to smaller post-projection perturbations.

avoids frequent oscillations between disparate sparsity patterns, which may be detrimental to the optimization process.

We operationalize this intuition through the use of a differentiable *soft top-$k$ masking* operation $\sigma_k^\beta : \mathbb{R}^d \to \mathbb{R}^d$. This function maps parameters $\theta$ to an approximately sparse output that suppresses low-magnitude entries in $\theta$. We parameterize the soft top-$k$ mask with a *sharpness parameter* $\beta$: at $\beta = 0$, $\sigma_k^\beta$ simply scales the input by a constant, and as $\beta \to \infty$, the mapping reduces to hard top-$k$ magnitude-based selection (Figure 1(a, b)). Soft top-$k$ masking therefore constrains the iterates $\theta_t$ to be close to the set of exactly $k$-sparse vectors, with the strength of this constraint mediated by the parameter $\beta$. Figures 1 (c) and 2 give some geometeric intuition for the effect of this mapping. We implement soft top-$k$ masking using a regularized optimal transportation formulation [11; 42] and demonstrate that this technique scales to networks on the order of $10^8$ parameters in size.

We evaluate Spartan on ResNet-50 [21] and ViT [15] models trained on the ImageNet-1K dataset. On ResNet-50, we find that sparse models trained with Spartan achieve higher generalization accuracies than those trained with existing methods at sparsity levels of 90% and above. In particular, we train ResNet-50 models to 76.5% top-1 validation accuracy at 95% sparsity and to 74.2% accuracy at 97.5% sparsity, improving on the previous state-of-the-art by 0.6% and 1.6% respectively. Our sparse ViT-B/16 models reduce model storage size by $10\times$ and inference FLOPs by $7.4\times$ at the cost of a 0.6% accuracy reduction relative to DeiT-B [38]. We further demonstrate that Spartan is effective for block structured pruning, a form of structured sparsity that is more amenable to acceleration on current GPU hardware than unstructured sparsity [18]. On a ViT-B/16 model with $16 \times 16$ block structured pruning, Spartan achieves 79.1% top-1 accuracy at 90% sparsity, compared to the baseline of 74.1% at the same sparsity level.

To summarize, we make the following contributions in this paper:

- We present Spartan, a sparsity-inducing training algorithm based on a soft top-$k$ masking operation. We show that Spartan interpolates between two existing sparse learning algorithms: iterative magnitude pruning [48] and Top-$K$ Always Sparse Training [24].

- We empirically evaluate Spartan using ResNet-50 and ViT models trained on the ImageNet-1K dataset, demonstrating consistent improvements over the prior state-of-the-art.

- We study the effect of Spartan's hyperparameters on an exploration-exploitation tradeoff during training and on the final accuracy of the trained models.

We provide an open source implementation of Spartan at `https://github.com/facebookresearch/spartan`.

**Notation.** We use $\mathbb{R}_+$ and $\mathbb{R}_{++}$ to denote the nonnegative real numbers and the strictly positive real numbers respectively. $\mathbb{1}_d$ is the all-ones vector of dimension $d$. Given a vector $x \in \mathbb{R}^d$, $\|x\|_p$ denotes

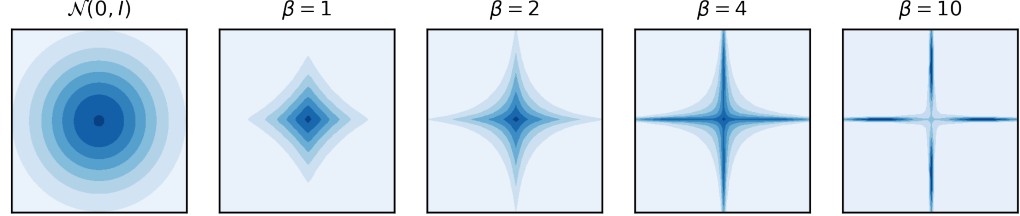

Figure 2: **A 2D example of soft masking.** We plot probability densities corresponding to the action of the soft top-1 operator $\sigma_1^\beta$ on the 2D standard Gaussian distribution (darker colors indicate higher density). Specifically, we visualize the densities of $\sigma_1^\beta(x) \mid x \sim \mathcal{N}(0, I_2)$ for a range of sharpness parameters $\beta$. Higher values of $\beta$ constrain iterates to be closer to the $k$-sparse feasible set.

the $p$-norm of $x$, $\|x\|_0$ is the number of nonzero entries in $x$, and $|x|$ is the vector of elementwise absolute values. When $x$ and $y$ are vectors, $x \odot y$ denotes elementwise multiplication and $x/y$ denotes elementwise division. The operator $\Pi_k(x) \coloneqq \arg\min_{y:\|y\|_0 \leq k} \|x - y\|_2$ denotes Euclidean projection onto the set of $k$-sparse vectors.

## 2   Related Work

**Neural network pruning.**  Our use of a magnitude-based criterion for sparse learning draws on early work in the area of neural network pruning [23]. Magnitude-based pruning is computationally cheap relative to alternative criteria that rely on first- or second-order information [33; 27; 20], and is a perhaps surprisingly performant option despite its simplicity [17]. More generally, Spartan builds on a substantial body of previous work that aims to jointly optimize sparsity patterns and model parameters for deep neural networks [43; 8; 19; 29; 28; 48, *inter alia*; see, *e.g.*, 22 for a survey].

**Spartan as a generalization of existing methods.**  We highlight two particularly relevant sparse training methods in the literature: the *iterative magnitude pruning* (IMP) method (Algorithm 1, [48]), and *Top-K Always Sparse Training*, or Top-KAST (Algorithm 2, [24]):

| **Algorithm 1** Iterative magnitude pruning update | **Algorithm 2** Dual averaging / Top-KAST update |
|---|---|
| **Input:** parameters $\theta_t$, loss function $L(\theta)$, sparsity budget $k$, step size $\eta_t$ | **Input:** parameters $\theta_t$, loss function $L(\theta)$, sparsity budget $k$, step size $\eta_t$ |
| **Output:** parameters $\theta_{t+1}$ | **Output:** parameters $\theta_{t+1}$ |
| 1: $\tilde{\theta}_t = \Pi_k(\theta_t)$ | 1: $\tilde{\theta}_t = \Pi_k(\theta_t)$ |
| 2: $\theta_{t+1} = \theta_t - \eta_t \nabla\tilde{\theta}_t \nabla L(\tilde{\theta}_t)$ | 2: $\theta_{t+1} = \theta_t - \eta_t \nabla L(\tilde{\theta}_t)$ |
| 3: **return** $\theta_{t+1}$ | 3: **return** $\theta_{t+1}$ |

Note that both Algorithms 1 and 2 compute the forward pass through the network using the sparsified parameters $\tilde{\theta}$, obtained by setting all but the top-$k$ entries of $\theta$ by magnitude to zero. Algorithms 1 and 2 differ in the presence of the $\nabla\tilde{\theta}_t$ term in line 2. This is a $d \times d$ diagonal matrix where $(\nabla\tilde{\theta}_t)_{ii} = 1$ iff $\theta_{t,i}$ was not set to zero by the projection $\Pi_k$, and $(\nabla\tilde{\theta}_t)_{ii} = 0$ otherwise.

At the extreme points of the sharpness parameter $\beta$, Spartan's parameter update procedure reduces to those of IMP and Top-KAST. We can thus view Spartan as a method that generalizes and smoothly interpolates between this pair of algorithms. As $\beta \to \infty$, Spartan is equivalent to IMP: this approach sparsifies parameters using a top-$k$ magnitude-based binary mask, and entries that were masked in the forward pass receive zero gradient in the backward pass. At $\beta = 0$, Spartan reduces to Top-KAST: this method again sparsifies parameters by magnitude with a binary mask, but unlike IMP, all entries are updated in the backward pass using the gradient of the loss with respect to the *masked* parameters. The Top-KAST update is thus an application of the *straight-through gradient method* [5; 9], otherwise known as *lazy projection* or *dual averaging* (DA) in optimization [34; 41; 2; 14]. See Sec. 5 for further discussion on this point.

**Smooth approximations.** Our use of the soft top-$k$ operation is related to prior methods that use the logistic sigmoid function as a differentiable approximation to the step function for sparse training [30; 36; 1]. These approaches similarly regulate the sharpness of the approximation with a temperature parameter that scales the input logits to the sigmoid function. A distinctive characteristic of Spartan is that the parameter $k$ directly controls the degree of sparsity of the mask; this differs from previous approaches involving the logistic sigmoid approximation that only indirectly control the degree of sparsity using an $L_1$ penalty term.

**Transformer-specific methods.** Several approaches are specialized to transformer-based architectures. These include structured pruning of pre-trained transformers for NLP tasks [39; 26; 40] and sparse training methods specifically applied to vision transformers [7; 6]. In contrast, Spartan is a general-purpose sparse training algorithm that is designed to be agnostic to the model architecture.

# 3 Sparsity via Regularized Transportation

We begin by motivating our proposed approach. A key advantage of the dual averaging method (Algorithm 2) over IMP (Algorithm 1) is that it mitigates the issue of gradient sparsity. In the IMP update, only those parameters that were not masked in the forward pass receive a nonzero gradient, resulting in slow progress at high levels of sparsity. In contrast, dual averaging applies a dense update to the parameter vector $\theta$ in each iteration—intuitively, these dense updates can help "revive" parameters that are below the top-$k$ magnitude threshold and are consequently masked to zero. Dual averaging iterates are thus able to more quickly explore the space of possible sparsity patterns.

On the other hand, large variations in the sparsity patterns realized post-projection can lead to instability in optimization, thus hampering the final performance of the model. For instance, Top-KAST empirically benefits from the application of additional sparsity-inducing regularization on the parameters [24 (Sec. 2.3), 37]. This issue motivates our use of a soft top-$k$ approximation as a mechanism for controlling the stability of our training iterates.

Each iteration of Spartan consists of the following two steps (Algorithm 3): (1) an approximately $k$-sparse masking of the model parameters using the soft top-$k$ operator, and (2) dual averaging-based parameter updates with the set of exactly $k$-sparse vectors as the feasible set. This procedure aims to combine the advantages of Algorithms 1 and 2 within a single parameter update scheme. Note that the Spartan parameter update is essentially identical to Algorithm 2, save for the inclusion of the intermediate soft top-$k$ masking step.

---

**Algorithm 3** Spartan parameter update

**Input:** parameters $\theta_t$, loss function $L(\theta)$, sparsity budget $k$, sharpness parameter $\beta$, step size $\eta_t$
**Output:** parameters $\theta_{t+1}$

1: $\sigma_k^\beta(\theta_t) = \theta_t \odot \mathrm{softtopk}(|\theta_t|, k, \beta)$         ▷ *apply soft masking*
2: $\tilde{\theta}_t = \Pi_k\left(\sigma_k^\beta(\theta_t)\right)$         ▷ *project onto $k$-sparse set*
3: $\theta_{t+1} = \theta_t - \eta_t \nabla \sigma_k^\beta(\theta_t) \nabla L(\tilde{\theta}_t)$         ▷ *compute dual averaging update*
4: **return** $\theta_{t+1}$

---

In the following, we begin by describing the soft top-$k$ masking operation and address the issues of scalability and of applying soft top-$k$ masking to induce structured sparsity. We then discuss how the update procedure outlined in Algorithm 3 can be incorporated within a complete training loop.

## 3.1 Soft Top-$k$ Masking

Our soft top-$k$ masking scheme is based on the soft top-$k$ operator $\mathrm{softtopk}(v, k, \beta)$ described by Xie et al. [42]. $\mathrm{softtopk}$ takes as input a vector $v \in \mathbb{R}^d$, a budget parameter $k > 0$ and a sharpness parameter $\beta \geq 0$, and outputs $m \in [0, 1]^d$, a smoothed version of the top-$k$ indicator vector. By using the magnitudes $|\theta|$ of the parameter vector as a measure of the "value" of each entry, we obtain a soft top-$k$ magnitude pruning operator $\sigma_k^\beta(\cdot)$ by masking the entries of the parameter vector with the output of the soft top-$k$ operator:

$$\sigma_k^\beta(\theta) := \theta \odot \mathrm{softtopk}(|\theta|, k, \beta).$$

---
**Algorithm 4** Soft top-$k$ forward pass
---
**Input:** values $v \in \mathbb{R}^d$, costs $c \in \mathbb{R}^d_{++}$, budget $k \in (0, \mathbb{1}^T_d c]$, sharpness parameter $\beta \geq 0$, max. iterations $T$, tolerance $\epsilon$, initial dual variable $\mu_0$
**Output:** mask $m \in [0,1]^d$, dual variables $\mu \in \mathbb{R}$, $\nu \in \mathbb{R}^d$
  1: $z = \beta v/c, \quad m_0 = \mathbb{1}_d$                          ▷ *normalize values $v$ by costs $c$*
  2: **for** $t = 1, \ldots, T$ **do**
  3:      $\nu_t = \log c - \log(\mathbb{1}_d + \exp(z + \mu_{t-1}))$          ▷ *normalize mask entries*
  4:      $\mu_t = \log k - \log \sum_i \exp(z_i + \nu_{t,i})$      ▷ *normalize sum to be equal to $k$*
  5:      $m_t = \exp(z + \mu_t + \nu_t - \log c)$                 ▷ *compute mask*
  6:      **if** $|v^T(m_t - m_{t-1})| < \epsilon|v^T m_{t-1}|$ **then**     ▷ *check stopping criterion*
  7:          **return** $m_t, \mu_t, \nu_t$
  8: **return** $m_T, \mu_T, \nu_T$

---
**Algorithm 5** Soft top-$k$ backward pass
---
**Input:** gradient w.r.t. outputs $g \in \mathbb{R}^d$, mask $m \in [0,1]^d$, costs $c$, budget $k$, sharpness parameter $\beta$
**Output:** gradient w.r.t. inputs
  1: $a_1 = \sum_i g_i m_i (1 - m_i), \quad a_2 = \sum_i c_i m_i^2$
  2: **return** $\beta m \odot (\mathbb{1}_d - m) \odot (g/c - a_1/(k - a_2))$

---

We introduce a mild generalization of the soft top-$k$ operator from Xie et al. [42] by incorporating a strictly positive *cost vector* $c$. In particular, we require that the output mask $m \in [0,1]^d$ satisfies the budget constraint $c^T m = k$. This abstraction is useful for modeling several notions of heterogeneous parameter costs within a model; for instance, parameters that are repeatedly invoked within a network have a higher associated computational cost. As a concrete example, Evci et al. [16] observed that the FLOP count of a ResNet-50 model at a fixed global sparsity can vary by a factor of over $2\times$ due to differences in the individual sparsities of convolutional layers with differing output dimensions.

To derive this cost-sensitive soft top-$k$ operator, we begin by expressing the mask $m$ as the solution to the following linear program (LP):

$$\underset{m \in \mathbb{R}^d}{\text{maximize}} \quad v^T m \quad \text{subject to} \quad 0 \preceq m \preceq \mathbb{1}_d, \quad c^T m = k. \tag{1}$$

Deferring the derivation to the Appendix, we can rewrite this LP as the following optimal transportation (OT) problem:

$$\underset{Y \in \mathbb{R}^{d \times 2}_+}{\text{minimize}} \quad \sum_{ij} C_{ij} Y_{ij} \quad \text{subject to} \quad Y \mathbb{1}_2 = c, \quad \mathbb{1}_d Y = [k, \mathbb{1}^T_d c - k], \tag{2}$$

with the cost matrix $C := [-v/c, 0]$. We can recover the mask from $Y$ as $m_i = Y_{i1}/c_i$. Now following [42; 11], we introduce entropic regularization to obtain a smoothed version of the top-$k$ operator, with the degree of smoothing controlled by $\beta$:

$$\underset{Y \in \mathbb{R}^{d \times 2}_+}{\text{minimize}} \quad \sum_{ij} C_{ij} Y_{ij} - \frac{1}{\beta} H(Y) \tag{3}$$

$$\text{subject to} \quad Y \mathbb{1}_2 = c, \quad \mathbb{1}_d Y = [k, \mathbb{1}^T_d c - k],$$

where $H(Y) := -\sum_{ij} Y_{ij}(\log Y_{ij} - 1)$ is the entropy of $Y$. In this standard form, we can efficiently solve this regularized OT problem using the Sinkhorn-Knopp algorithm [11; 4; 12].

Algorithms 4 and 5 describe the forward and backward passes of the resulting cost-sensitive soft top-$k$ operator (see Appendices B and C for their derivations). The expression for the gradient in Algorithm 5 follows from Theorem 3 in Xie et al. [42] with some algebraic simplification. We remark that this closed-form backward pass is approximate in the sense that it assumes that we obtain the *optimal* dual variables $\mu, \nu$ in the forward pass; in practice, we do not encounter issues when using estimates of the dual variables obtained with tolerance $\epsilon = 0.01$ and a maximum iteration count of $T = 100$ in Algorithm 4.

**Scalability.** Since we apply soft top-$k$ masking to high-dimensional parameters in each iteration of training, it is important to minimize the computational overhead of Sinkhorn iteration in the

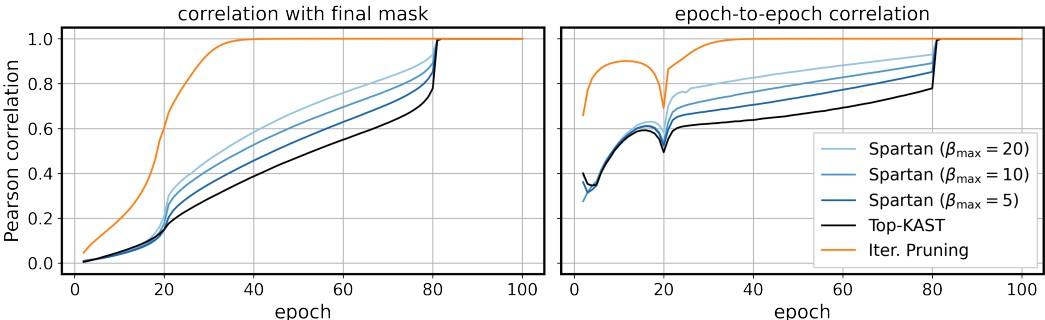

Figure 3: **Spartan parameterizes an exploration-exploitation tradeoff.** For each run of ResNet-50 training on ImageNet-1K, we plot Pearson correlation coefficients between the sparsity mask at the end of each epoch and the mask obtained at the end of training (left), and between sparsity masks at the end of subsequent epochs (right). Kinks in the correlation curves at epochs 20 and 80 are respectively due to the end of sparsity annealing and the start of fine-tuning with fixed masks (see Section 3.2 for details on the training schedule).

forward pass. We note that in order to compute the hard top-$k$ projection step in dual averaging (as in Algorithm 3), it is necessary to find the index $i_k$ of the $k$th largest entry of $|\theta|/c$. Given the index $i_k$, we can use the heuristic initialization $\mu_0 = -\beta|\theta_{i_k}|/c_{i_k}$ for the dual variable $\mu$ to accelerate the convergence of Sinkhorn iteration. This heuristic is based on the observation that $|\theta_{i_k}|/c_{i_k}$ is the threshold value of the normalized value vector $|\theta|/c$ in *hard* top-$k$ masking. Concretely, we demonstrate in Sec. 4.3 that our implementation of Spartan incurs a per-iteration runtime overhead of approximately 5% over standard dense ResNet-50 training.

**Structured sparsity.** To implement block structured sparsity, we instantiate one mask variable per block and mask all parameters within that block with the same mask variable. To compute the mask, we use the sum of the magnitudes of the entries in each block as the corresponding value $v_i$. In a standard pruning setting with uniform costs across all parameters, the corresponding cost $c_i$ is simply the total number of entries in the block.

## 3.2 Training with Spartan

**Training schedule.** When training with Spartan, we divide the training process into a warmup phase, an intermediate phase, and a fine-tuning phase. In our experiments, the warmup phase consists of the first 20% of epochs, the intermediate phase the next 60%, and the fine-tuning phase the final 20%. In the warmup phase, we linearly anneal the global sparsity of the model from a value of 1 at the start of training to the target level of sparsity. Throughout the intermediate phase, we maintain the target sparsity level, and during the fine-tuning phase, we fix the sparsity mask to that used at the start of fine-tuning. This training schedule is similar to those used in prior work [37].

**Exploration vs. exploitation as a function of $\beta$.** From the start of training up to the start of the fine-tuning phase, we linearly ramp the sharpness parameter $\beta$ from an initial value of 1 to the final value $\beta_{\max}$. We interpret the spectrum of updates parameterized by $\beta$ as being characteristic of an exploration-exploitation tradeoff. In Figure 3, we illustrate this phenomenon by plotting Pearson correlation coefficients between sparsity masks at different stages of training. We observe that iterative magnitude pruning converges on the final mask relatively early in training, which is indicative of insufficient exploration of different sparsity patterns—consequently, this model achieves lower validation accuracy than the remaining models. The three models trained with Spartan each ramp $\beta$ from 1 at the start of training to $\beta_{\max}$ at epoch 80. The value of $\beta_{\max}$ correlates well both with the Pearson correlation with the final mask, and with the Pearson correlation between masks at the end of subsequent epochs. Spartan thus interpolates between the high-exploration regime of standard dual averaging and the low-exploration regime of iterative magnitude pruning. In this example, the intermediate setting $\beta_{\max} = 10$ achieves the highest top-1 validation accuracy of 74.6%, with Top-KAST at 73.5% and IMP at 68.2%.

Table 1: Top-1 accuracies on ImageNet-1K validation set with fully dense training.

| Method | Epochs | Accuracy (%) | Method | Epochs | Accuracy (%) |
|---|---|---|---|---|---|
| ResNet-50 | 100 | $77.08 \pm 0.11$ | ViT-B/16 | 100 | $77.78 \pm 0.22$ |
| | 200 | $\mathbf{77.46} \pm 0.09$ | | 300 | $\mathbf{80.06} \pm 0.11$ |
| | 400 | $77.17 \pm 0.04$ | | | |

Table 2: ResNet-50 top-1 accuracies on ImageNet-1K validation set at varying levels of sparsity and epochs of training. When available, we report means and standard deviations over 3 trials.

| | Method | Epochs | Sparsity 80% | 90% | 95% | 97.5% | 99% |
|---|---|---|---|---|---|---|---|
| [16] | RigL (ERK) | 100 | $75.1 \pm 0.05$ | $73.0 \pm 0.04$ | $69.7 \pm 0.17$ | - | - |
| | | 500 | $77.1 \pm 0.06$ | $76.4 \pm 0.05$ | $74.5 \pm 0.09$ | - | - |
| [25] | STR[†] | 100 | 76.19 | 74.31 | 70.40 | 62.84 | 51.82 |
| [47] | ProbMask | 100 | - | 74.68 | 71.50 | 66.83[‡] | 61.07 |
| [46] | OptG | 100 | - | 74.28 | 72.38 | - | 62.10 |
| | IMP | 100 | $75.3 \pm 0.07$ | $73.7 \pm 0.14$ | $70.6 \pm 0.05$ | - | - |
| | Top-KAST | 100 | $76.08 \pm 0.02$ | $75.13 \pm 0.03$ | $73.19 \pm 0.02$ | - | - |
| | (with PP) | 100 | $76.24 \pm 0.07$ | $75.23 \pm 0.02$ | $73.25 \pm 0.02$ | - | - |
| [37] | (with ERK) | 100 | $76.42 \pm 0.03$ | $75.51 \pm 0.05$ | - | - | - |
| | (with PP | 100 | $76.76 \pm 0.08$ | $75.74 \pm 0.08$ | - | - | - |
| | & ERK) | 200 | $77.51 \pm 0.03$ | $76.94 \pm 0.10$ | - | - | - |
| | | 300 | $77.64 \pm 0.05$ | $77.16 \pm 0.19$ | - | - | - |
| | Top-KAST | 100 | $76.66 \pm 0.04$ | $75.48 \pm 0.15$ | $73.51 \pm 0.16$ | $70.23 \pm 0.05$ | - |
| | | 200 | $77.55 \pm 0.08$ | $76.84 \pm 0.11$ | $75.20 \pm 0.11$ | $71.72 \pm 0.06$ | - |
| | | 400 | $\mathbf{77.75} \pm 0.09$ | $77.37 \pm 0.07$ | $75.90 \pm 0.04$ | $72.58 \pm 0.13$ | - |
| | Spartan | 100 | $76.89 \pm 0.09$ | $\mathbf{76.17} \pm 0.10$ | $74.68 \pm 0.24$ | $71.95 \pm 0.11$ | $\mathbf{63.87} \pm 0.12$ |
| | | 200 | $77.56 \pm 0.19$ | $77.06 \pm 0.13$ | $75.92 \pm 0.01$ | $73.41 \pm 0.15$ | $65.77 \pm 0.03$ |
| | | 400 | $77.57 \pm 0.08$ | $77.40 \pm 0.06$ | $76.48 \pm 0.20$ | $74.15 \pm 0.10$ | $66.80 \pm 0.03$ |

[†]Reported accuracies for models closest to the listed sparsity level. [‡]Model trained at 98% sparsity.

# 4  Empirical Evaluation

In this section, we report the results of our sparse training experiments on the ImageNet-1K dataset with two standard architectures: ResNet-50 [21] and ViT-B/16 [15], consisting of 25.6M and 86.4M parameters respectively. On ResNet-50, we evaluate only unstructured pruning, whereas on ViT-B/16, we evaluate both unstructured and block structured pruning. We subsequently present empirical studies of the sensitivity of Spartan to the value of $\beta$, the effect of running Spartan without the dual averaging step, and the computational overhead of soft top-$k$ masking.

## 4.1  ImageNet-1K Classification

In all our experiments, we run Spartan with the training schedule described in Section 3.2. We train and evaluate our models on the ImageNet-1K dataset with the standard training-validation split and report means and standard deviations over 3 independent trials. We provide additional details on our experimental setup in the Appendix.

**ResNet-50 experimental setup.** For consistency with our baselines, we use standard data augmentation with horizontal flips and random crops at $224 \times 224$ resolution. For all Spartan runs, we use $\beta_{\max} = 10$, which we selected based on models trained at 95% accuracy. We sparsify the parameters of all linear and convolutional layers with a global sparsity budget, excluding bias parameters and the parameters of batch normalization layers. Our baselines are iterative magnitude pruning [48], RigL with the Erdos-Renyi-Kernel (ERK) sparsity distribution [16], Soft Threshold Weight Reparameterization (STR) [25], probabilistic masking (ProbMask) [47], OptG [46], and Top-KAST with Powerpropagation and ERK [24; 37]. We additionally re-run the most performant baseline method,

Table 3: ViT-B/16 top-1 accuracies on ImageNet-1K validation set at 90% sparsity with unstructured sparsity and block structured pruning of attention layers.

| Method | Epochs | Sparsity Structure | | |
|---|---|---|---|---|
| | | Unstructured | $16 \times 16$ blocks | $32 \times 32$ blocks |
| Top-KAST | 100 | $78.05 \pm 0.07$ | $67.69 \pm 0.23$ | $64.79 \pm 0.14$ |
| | 300 | $80.86 \pm 0.03$ | $74.11 \pm 0.20$ | $70.67 \pm 0.96$ |
| Spartan | 100 | $\mathbf{79.88} \pm 0.16$ | $\mathbf{75.62} \pm 0.07$ | $\mathbf{74.50} \pm 0.23$ |
| | 300 | $\mathbf{81.18} \pm 0.04$ | $\mathbf{79.12} \pm 0.16$ | $\mathbf{78.43} \pm 0.10$ |

Top-KAST, using a reimplementation in our own codebase. For Top-KAST, we exclude the first convolutional layer from pruning (following [37; 16]) and we use fully dense backward passes (*i.e.*, with a backward sparsity of 0), since this setting achieved the highest accuracy in prior work [24]. We use mixed precision training with a batch size of 4096 on 8 NVIDIA A100 GPUs.

**ViT experimental setup.** We use the ViT-B architecture with $16 \times 16$ patches at $224 \times 224$ input resolution. We augment the training data using RandAugment [10], MixUp [45] and CutMix [44]. Our ViT models are trained from random initialization, without any pretraining. We set $\beta_{\max} = 20$ for Spartan with unstructured sparsity, and $\beta_{\max} = 320$ and $\beta_{\max} = 640$ for Spartan with $16 \times 16$ and $32 \times 32$ blocks respectively. These latter settings are the values of $\beta_{\max}$ for the unstructured case scaled up by factors of 16 and 32—since each block averages the magnitudes of $B^2$ entries, we expect the variance of the block values to correspondingly decrease by a factor of $B^2$, and we thus compensate by scaling up $\beta$ by $B$. In the block structured case, we exclude the input convolutional layer and the output classification head from pruning since their parameter dimensions are not divisible by the block size. We use mixed precision training with a batch size of 4096 on 16 NVIDIA A100 GPUs across 2 nodes.

**Results.** Table 1 lists the top-1 validation accuracies achieved by fully dense ResNet-50 and ViT-B/16 models. Table 2 reports validation accuracies for ResNet-50 models at 80%, 90%, 95%, 97.5% and 99% sparsity, and Table 3 reports validation accuracies for ViT at 90% sparsity. In the Appendix, we report additional measurements of inference FLOP costs for our sparse models and the results of experiments with FLOP-sensitive pruning.

For ResNet-50 models, we find that Spartan outperforms all our baselines across all training durations at sparsity levels of 90% and above. In particular, Spartan achieves a mean top-1 accuracy within 1% of fully dense training at 95% sparsity. We observe that additional epochs of sparse training consistently improves the final generalization accuracy; in contrast, validation accuracy peaks at 200 epochs for dense training. This trend persists at 800 training epochs, where Spartan achieves $67.18 \pm 0.13$ top-1 accuracy at 99% sparsity. For the Top-KAST baseline, we omit results at 99% sparsity due to training instability. We note that the accuracy improvements in the Spartan-trained ResNet-50 models relative to Top-KAST are not a result of increased FLOP counts at a given level of sparsity, as can be seen from the FLOP measurements reported in Appendix E.

For ViT-B/16 models, Spartan outperforms Top-KAST for both unstructured and block structured pruning. We observe a particularly large improvement over Top-KAST in the block structured case, where Spartan improves absolute validation accuracy by 7.8% for $32 \times 32$ block structured pruning. For unstructured pruning, Spartan achieves comparable accuracy to SViTE [7] (81.2% for Spartan vs. 81.3% for SViTE), but with 30% higher sparsity (90% for Spartan vs. 60% for SViTE). In exchange for a 0.6% reduction in accuracy relative to DeiT-B [38], Spartan reduces model storage cost by $10\times$ and the FLOP cost of inference by $7.4\times$.

## 4.2 Sensitivity and Ablation Analysis

Figure 4 (left) shows the effect of ablating the dual averaging step in Spartan—*i.e.*, omitting hard top-$k$ projection in the forward pass—over a range of settings of $\beta_{\max}$ for 95% sparse ResNet-50 models trained for 100 epochs. The dashed line shows top-1 accuracy with Top-KAST. For Spartan training without dual averaging, we compute a hard top-$k$ mask at the end of epoch 80, and as with standard Spartan training, we fix this mask until the end of training. In the absence of top-$k$ projection, we find that accuracy increases with increasing $\beta_{\max}$ up to $\beta_{\max} = 80$; at lower settings of $\beta_{\max}$,

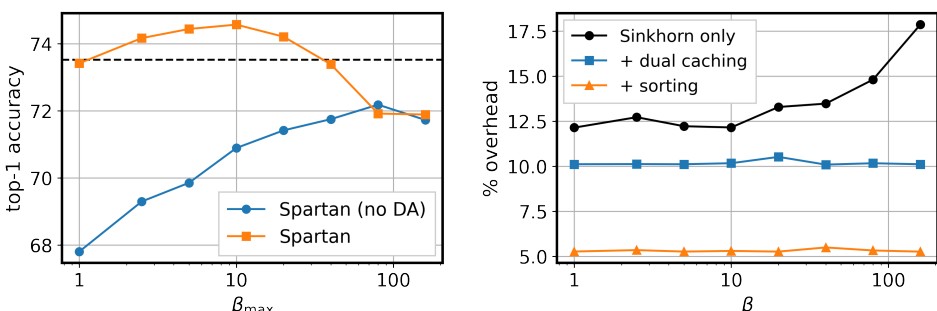

Figure 4: **Effect of $\beta$ and dual averaging (left).** Top-1 ImageNet-1K validation accuracies for Spartan with and without dual averaging (DA) and with varying $\beta_{\max}$ at 95% sparsity. **Computational overhead (right).** Percentage increase in per-iteration wall clock runtime over dense training for Spartan with standard Sinkhorn iteration, Sinkhorn with dual caching, and Sinkhorn with sorting.

the higher approximation error of soft masking is detrimental to the final accuracy of the model. In contrast, the use of top-$k$ projection in the forward pass mitigates this mismatch between training and inference and improves the final accuracy of the sparse model.

### 4.3 Computational Overhead

We evaluate the computational overhead of Spartan over standard dense training by measuring wall clock training times for a ResNet-50 model (Figure 4, right). Our benchmark measures runtime on a single NVIDIA A100 GPU over 50 iterations of training with a batch size of 256. We use random inputs and labels to avoid incurring data movement costs. We compare three approaches: standard Sinkhorn iteration, Sinkhorn iteration with the dual variable $\mu$ initialized using its final value from the previous iteration ("dual caching"), and Sinkhorn iteration with $\mu$ initialized using the value $-\beta|\theta_{i_k}|/c_{i_k}$, where we compute the index $i_k$ of the $k$th largest entry of the normalized value vector $|\theta|/c$ by sorting its entries in each iteration ("sorting").

Standard Sinkhorn iteration incurs higher overheads as $\beta$ increases—this is due to the additional iterations required to reach convergence as the regularized OT problem more closely approximates the original LP. We find that dual caching and sorting both prevent this growth in runtime over the range of $\beta$ values that we tested. In our remaining experiments, we use the simple sorting-based approach since it corresponds to the lowest relative overhead over standard dense training (approximately 5%). We note that this relative overhead decreases as the batch size increases since the cost of computing the mask is independent of the batch size. Spartan is therefore best suited to training with large batch sizes since we amortize the cost of mask updates over the size of the batch.

## 5 Discussion

**Extensions.** As an alternative to magnitude-based pruning, we may also apply Spartan in conjunction with *learned value parameters*, as in methods such as Movement Pruning [35]. In this approach, we would compute masks using a set of auxiliary parameters $\phi \in \mathbb{R}^d$ instead of the magnitudes $|\theta|$: $\sigma_k^\beta(\theta;\phi) = \theta \odot \mathrm{softtopk}(\phi, k, \beta)$, and similarly for hard projection. We remark that while this variant requires additional memory during training to store the value parameters, there is no additional cost during inference since the sparsity mask is fixed at the end of training.

**Limitations.** Since Spartan retains a dense parameter vector and computes dense backward passes during training, it incurs higher memory and computational costs in each iteration than methods like RigL [16] that use both a sparse parameters and sparse backward passes. Nevertheless, we note that in terms of *total* computational or energy cost over a full training run, Spartan may remain a competitive option as it requires fewer iterations of training to reach a given accuracy threshold relative to these sparse-to-sparse methods. However, we do not compare total training FLOP costs in our empirical evaluation.

A further limitation is that cost-sensitive pruning with Spartan is only compatible with relatively crude linear cost models of the form $c^T m$, where $c$ is the cost vector and $m$ is the mask. This restriction

arises due to the requirements of the regularized OT formulation used to compute the soft top-$k$ mask. In particular, this limitation precludes the use of cost models involving interaction terms such as those arising from spatially coherent sparsity patterns. For example, the cost models that we consider here cannot encode the notion that coherently pruning an entire row or column of a parameter matrix will yield more cost savings than pruning random entries of the matrix.

**Optimization with Dual Averaging.** As discussed in Section 3, Spartan and Top-KAST are both applications of the dual averaging method [34; 41]. The empirically observed effectiveness of dual averaging in sparse training is reminiscent of its success in a related domain where continuous optimization is complicated by discrete constraints: neural network quantization. In this area, the BinaryConnect algorithm [9], which trains binary neural networks with parameters in $\{\pm 1\}^d$, is considered to be a foundational method. As observed by Bai et al. [2], BinaryConnect is itself a special case of dual averaging, and the theoretical underpinnings of this connection to dual averaging were further analyzed by Dockhorn et al. [14]. These parallels with neural network quantization suggest that similar ideas may well apply towards deepening our understanding of sparse learning.

**Societal Impacts.** Inference with deep neural network models is a computationally intensive process. At present, the total energy footprint associated with serving these models in production is expected to continue growing in tandem with the rising prevalence of large transformer-based architectures in vision and NLP applications. Research towards improving the energy efficiency of deep neural networks is therefore an important counterbalancing force against increasing resource usage by these models. The development of sparse learning algorithms is particularly relevant to these efforts, and we expect that the impact of these approaches will further increase as sparsity-aware hardware acceleration becomes more widely available.

## 6  Conclusions & Future Work

In this work, we describe a sparse learning algorithm that interpolates between two parameter update schemes: standard stochastic gradient updates with hard masking and the dual averaging method. We show that there exists an intermediate regime between these two methods that yields improved generalization accuracy for sparse convolutional and transformer models, particularly at higher levels of sparsity. While we have demonstrated promising empirical results with our proposed method, the learning dynamics of stochastic optimization for deep networks under sparsity constraints remains relatively poorly understood from a theoretical standpoint. There thus exists ample opportunity for further work towards better understanding sparse learning algorithms, which may in turn inspire future algorithmic advances in this area.

## Acknowledgements and Disclosure of Funding

We are grateful to Trevor Gale for his feedback on this work. We also thank our anonymous reviewers for their valuable suggestions on improving the manuscript. This work was funded by Meta.

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
