| | | 500 | 77.1 ±0.06 | 76.4 ±0.05 | 74.5 ±0.09 | - | - |
| [25] | STR[†] | 100 | 76.19 | 74.31 | 70.40 | 62.84 | 51.82 |
| [47] | ProbMask | 100 | - | 74.68 | 71.50 | 66.83[‡] | 61.07 |
| [46] | OptG | 100 | - | 74.28 | 72.38 | - | 62.10 |
| | IMP | 100 | 75.3 ±0.07 | 73.7 ±0.14 | 70.6 ±0.05 | - | - |
| | Top-KAST | 100 | 76.08 ±0.02 | 75.13 ±0.03 | 73.19 ±0.02 | - | - |
| | (with PP) | 100 | 76.24 ±0.07 | 75.23 ±0.02 | 73.25 ±0.02 | - | - |
| [37] | (with ERK) | 100 | 76.42 ±0.03 | 75.51 ±0.05 | - | - | - |
| | (with PP | 100 | 76.76 ±0.08 | 75.74 ±0.08 | - | - | - |
| | & ERK) | 200 | 77.51 ±0.03 | 76.94 ±0.10 | - | - | - |
| | | 300 | 77.64 ±0.05 | 77.16 ±0.19 | - | - | - |
| | Top-KAST | 100 | 76.66 ±0.04 | 75.48 ±0.15 | 73.51 ±0.16 | 70.23 ±0.05 | - |
| | | 200 | 77.55 ±0.08 | 76.84 ±0.11 | 75.20 ±0.11 | 71.72 ±0.06 | - |
| | | 400 | **77.75** ±0.09 | 77.37 ±0.07 | 75.90 ±0.04 | 72.58 ±0.13 | - |
| | Spartan | 100 | 76.89 ±0.09 | **76.17** ±0.10 | 74.68 ±0.24 | 71.95 ±0.11 | **63.87** ±0.12 |
| | | 200 | **77.56** ±0.19 | **77.06** ±0.13 | 75.92 ±0.01 | **73.41** ±0.15 | 65.77 ±0.03 |
| | | 400 | 77.57 ±0.08 | **77.40** ±0.06 | 76.48 ±0.20 | 74.15 ±0.10 | 66.80 ±0.03 |

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

## A Derivation of the Optimal Transport LP (Problem 2)

Here, we show how the original top-$k$ LP with costs $c$ can be straightforwardly rewritten in the form of an optimal transportation problem. For a given value vector $v \in \mathbb{R}^d$, cost vector $c \in \mathbb{R}^d_{++}$, and budget $k$, the top-$k$ LP is:

$$\underset{m \in \mathbb{R}^d}{\text{maximize}} \quad v^T m$$
$$\text{subject to} \quad 0 \preceq m \preceq \mathbb{1}_d,$$
$$c^T m = k.$$

Define $y := c \odot m$ and substitute to obtain:

$$\underset{y \in \mathbb{R}^d}{\text{maximize}} \quad (v/c)^T y$$
$$\text{subject to} \quad 0 \preceq y \preceq c,$$
$$\mathbb{1}_d^T y = k.$$

Now eliminate the upper bound constraint by introducing additional variables $y'$ to give:

$$\underset{y, y' \in \mathbb{R}^d}{\text{maximize}} \quad (v/c)^T y$$
$$\text{subject to} \quad y \succeq 0, \; y' \succeq 0,$$
$$y + y' = c,$$
$$\mathbb{1}_d^T y = k.$$

Finally, we rewrite this problem in the form of a standard OT problem by introducing the variable $Y := [y, \, y']$ and the cost matrix $C := [-v/c, \, 0]$ to yield:

$$\underset{Y \in \mathbb{R}_+^{d \times 2}}{\text{minimize}} \quad \sum_{ij} C_{ij} Y_{ij}$$
$$\text{subject to} \quad Y \mathbb{1}_2 = c,$$
$$\mathbb{1}_d Y = [k, \, \mathbb{1}_d^T c - k],$$

where we obtain the second column constraint by combining $y + y' = c$ and $\mathbb{1}_d^T y = k$.

## B Derivation of Soft Top-$k$ Forward Pass (Algorithm 4)

Problem 3 is an entropy regularized optimal transport problem with cost matrix $C := [-v/c, 0] \in \mathbb{R}^{d \times 2}$, row marginals $c \in \mathbb{R}^d$, and column marginals $[k, \mathbb{1}_d^T c - k]$. By Lemma 2 in [11], the optimal solution to this problem can be written in the form $\text{diag}(\exp \nu) \exp(-\beta C) \text{diag}(\exp[\mu, \mu'])$, with dual variables $\mu, \mu' \in \mathbb{R}$ and $\nu \in \mathbb{R}^d$. Moreover, we can compute a sequence of iterates converging to an optimal collection of dual variables using Sinkhorn iteration.

Note that the dual variables $\mu, \mu'$, and $\nu$ are only unique up to an additive constant: for any $\mu, \mu', \nu$, and $\delta$, we have $\text{diag}(\exp \nu) \exp(-\beta C) \text{diag}(\exp[\mu, \mu']) = \text{diag}(\exp(\nu + \delta)) \exp(-\beta C) \text{diag}(\exp[\mu - \delta, \mu' - \delta])$. We use this degree of freedom to fix $\mu' = 0$.

This yields the following Sinkhorn updates:

$$\mu_{t+1} = \log k - \log \sum_{i=1}^d \exp(-\beta C_{i1} + \nu_{t,i})$$
$$= \log k - \log \sum_{i=1}^d \exp(\beta v_i / c_i + \nu_{t,i}),$$
$$\nu_{t+1} = \log c - \log(\exp(-\beta C_{\cdot 1} + \mu_t) + \exp(-\beta C_{\cdot 2} + \mu_t'))$$
$$= \log c - \log(\exp(\beta v / c + \mu_t) + \mathbb{1}_d),$$

as stated in Algorithm 4. Note that eliminating $\mu'$ reduces the cost of computing the forward pass by roughly $1/3$.

# C Derivation of Soft Top-$k$ Backward Pass (Algorithm 5)

The gradient of the loss with respect to the input values $v$ of the soft top-$k$ function follows from Theorem 3 of [42] with some algebraic manipulation. We restate the theorem below with the notation used in this paper. In the following, let $\bar{q} := q_{:-1}$ denote the vector $q$ with the last entry removed, and let $\bar{Y} := Y_{.,:-1}$ denote the matrix $Y$ with the last column removed.

---

**Theorem [42].** Let the solution $Y \in \mathbb{R}^{N \times M}$ of the entropy regularized optimal transport problem with cost matrix $C$, row marginals $p$ and column marginals $q$ be given by:

$$Y_{ij} = \exp(-\beta C_{ij} + \nu_i^* + \mu_j^*),$$

where $\nu^* \in \mathbb{R}^N$, and $\mu^* \in \mathbb{R}^M$ are optimal dual variables.
Then $\frac{d\nu^*}{dC}$ and $\frac{d\mu^*}{dC}$ are given by:

$$\begin{bmatrix} \frac{d\nu^*}{dC} \\ \frac{d\mu^*}{dC} \end{bmatrix} = \begin{bmatrix} -H^{-1}D \\ \mathbf{0} \end{bmatrix},$$

where $-H^{-1}D \in \mathbb{R}^{(N+M-1) \times N \times M}$, $\mathbf{0} \in \mathbb{R}^{1 \times N \times M}$, and

$$D_{\ell i j} = \beta \begin{cases} \delta_{\ell i} Y_{ij}, & \ell = 1, \ldots, N \\ \delta_{(\ell-N)j} Y_{ij}, & \ell = N+1, \ldots, N+M-1 \end{cases},$$

$$H^{-1} = -\begin{bmatrix} (\text{diag}(p))^{-1} + (\text{diag}(p))^{-1}\bar{Y}\mathcal{K}^{-1}\bar{Y}^T(\text{diag}(p))^{-1} & -(\text{diag}(p))^{-1}\bar{Y}\mathcal{K}^{-1} \\ -\mathcal{K}^{-1}\bar{Y}^T(\text{diag}(p))^{-1} & \mathcal{K}^{-1} \end{bmatrix},$$

$$\mathcal{K} = \text{diag}(\bar{q}) - \bar{Y}^T(\text{diag}(p))^{-1}\bar{Y}.$$

---

Specializing this theorem to our setting with $N = d$ and $M = 2$, and recalling that the soft top-$k$ mask values are given by $m_i = Y_{i1}/c_i$, we obtain:

$$H^{-1} = \frac{1}{\mathcal{K}}\begin{bmatrix} -\mathcal{K}\text{diag}(1/c) - mm^T & m \\ m^T & -1 \end{bmatrix},$$

$$\mathcal{K} = k - \mathbb{1}^T(m^2 \odot c).$$

Therefore:

$$\begin{bmatrix} \frac{d\nu^*}{dC_{.1}} \\ \frac{d\mu^*}{dC_{.1}} \end{bmatrix} = \frac{\beta}{\mathcal{K}}\begin{bmatrix} \mathcal{K}\text{diag}(m) + m(m^2 \odot c)^T - m(m \odot c)^T \\ -(m^2 \odot c)^T + (m \odot c)^T \end{bmatrix}.$$

Taking the derivative of the mask with respect to the first column of the cost matrix, we have:

$$\frac{dm}{dC_{.1}} = -\beta\text{diag}(m) + m\frac{d\mu^*}{dC_{.1}} + \text{diag}(m)\frac{d\nu^*}{dC_{.1}}$$

$$= -\beta\left(\text{diag}(m \odot (\mathbb{1} - m)) - \frac{1}{\mathcal{K}}(m \odot (\mathbb{1} - m))(m \odot (\mathbb{1} - m) \odot c)^T\right).$$

Finally, using our definition of the cost matrix as $C_{.1} = -v/c$ and applying the loss gradient $g := \left(\frac{dL}{dm}\right)^T$, we obtain:

$$\left(\frac{dL}{dv}\right)^T = \left(\frac{dL}{dm}\frac{dm}{dv}\right)^T = \beta m \odot (\mathbb{1} - m) \odot \left(\frac{g}{c} - \frac{1}{\mathcal{K}}\mathbb{1}^T(g \odot m \odot (\mathbb{1} - m))\right),$$

as desired.

# D Training Details

In Tables 4 and 5, we detail the hyperparameters used for our training runs. These hyperparameters were derived from `https://github.com/pytorch/vision/blob/96d1fecf/references/classification/README.md`.

Table 4: ResNet-50 training hyperparameters

| Hyperparameter | Value |
|---:|:---|
| optimizer | Nesterov accelerated gradient method ($\mathrm{momentum} = 0.9$) |
| max. learning rate | 1.0 |
| min. learning rate | 0.001 |
| learning rate warmup epochs | 5 |
| learning rate decay schedule | cosine |
| batch size | 4096 |
| weight decay | $10^{-4}$ (0.0 for bias and normalization parameters) |
| label smoothing | 0.1 |
| data augmentation | random crops, random horizontal flips |
| input resolution | $224 \times 224$ |
| sparsity annealing schedule | linear from 1 to target sparsity at epoch fraction 0.2 |
| $\beta$ annealing schedule | linear from 1 to $\beta_{\max}$ at epoch fraction 0.8 |
| Sinkhorn max. iterations | 100 |
| Sinkhorn tolerance $\epsilon$ | 0.01 |

Table 5: ViT-B/16 training hyperparameters

| Hyperparameter | Value |
|---:|:---|
| optimizer | AdamW ($\beta = (0.9, 0.999), \epsilon = 10^{-8}$) |
| max. learning rate | 0.003 |
| min. learning rate | 0.0 |
| learning rate warmup epochs | 10% of total epochs |
| learning rate decay schedule | cosine |
| batch size | 4096 |
| weight decay | 0.3 (0.0 for bias and normalization parameters) |
| label smoothing | 0.1 |
| data augmentation | random crops, random horizontal flips, RandAugment (ops = 2, magnitude = 9) |
| mixup $\alpha$ | 0.2 |
| CutMix $\alpha$ | 1.0 |
| gradient $L_2$ norm clip | 1.0 |
| input resolution | $224 \times 224$ |
| exponential moving averaging | false |
| sparsity annealing schedule | linear from 1 to target sparsity at epoch fraction 0.2 |
| $\beta$ annealing schedule | linear from 1 to $\beta_{\max}$ at epoch fraction 0.8 |
| Sinkhorn max. iterations | 100 |
| Sinkhorn tolerance $\epsilon$ | 0.01 |

# E FLOP Measurements

Due to differences in the computational cost associated with individual parameters, the sparsity fraction does not map 1-to-1 to the fraction of FLOPs required for inference. Tables 6 and 7 give FLOP costs for our sparse models as a percentage of the FLOP cost of the corresponding dense model. We performed our FLOP measurements using the open source tool available at `https://github.com/sovrasov/flops-counter.pytorch/`.

We count multiply and add operations as one FLOP each. We remark that there exists some inconsistency in the literature regarding this convention, with some prior work using multiply-accumulate (MAC) counts and FLOP counts interchangeably. To convert the base FLOP counts listed below for ResNet-50 and ViT-B/16 to MACs, we can simply divide the given counts by 2.

In Table 6, the ResNet-50 FLOP counts for Top-KAST are slightly higher than those for Spartan, possibly due to the exclusion of the input convolutional layer from pruning in the case of Top-KAST.[1] In particular, this demonstrates that the accuracy improvements seen in Spartan-trained models over those trained using Top-KAST do not correlate with an increase in their FLOP costs.

Table 6: Percentage FLOP costs of sparse ResNet-50 models relative to the FLOP cost of a dense ResNet-50. The cost of a dense ResNet-50 model is 8.24 GFLOPs.

| Method | Epochs | Sparsity | | | | |
|---|---|---|---|---|---|---|
| | | **80%** | **90%** | **95%** | **97.5%** | **99%** |
| Top-KAST | 100 | $28.43 \pm 0.28$ | $17.11 \pm 0.88$ | $11.21 \pm 0.29$ | $7.82 \pm 0.08$ | - |
| | 200 | $27.80 \pm 0.31$ | $17.66 \pm 0.07$ | $11.75 \pm 0.12$ | $7.93 \pm 0.13$ | - |
| | 400 | $28.06 \pm 0.12$ | $18.05 \pm 0.28$ | $11.40 \pm 0.16$ | $7.74 \pm 0.19$ | - |
| Spartan | 100 | $24.68 \pm 0.07$ | $14.48 \pm 0.11$ | $8.67 \pm 0.14$ | $5.04 \pm 0.11$ | $2.53 \pm 0.04$ |
| | 200 | $24.37 \pm 0.60$ | $14.37 \pm 0.09$ | $8.43 \pm 0.06$ | $4.97 \pm 0.07$ | $2.59 \pm 0.04$ |
| | 400 | $23.97 \pm 0.31$ | $14.20 \pm 0.08$ | $8.56 \pm 0.10$ | $5.07 \pm 0.06$ | $2.50 \pm 0.03$ |

Table 7: Percentage FLOP costs of 90% sparse ViT-B/16 models at $224 \times 224$ input resolution relative to the FLOP cost of a dense ViT-B/16 model. The cost of a dense ViT-B/16 model is 35.19 GFLOPs.

| Method | Epochs | Sparsity Structure | | |
|---|---|---|---|---|
| | | **Unstructured** | **$16 \times 16$ blocks** | **$32 \times 32$ blocks** |
| Top-KAST | 100 | $13.42 \pm 0.00$ | $12.76 \pm 0.00$ | $12.76 \pm 0.00$ |
| | 300 | $13.41 \pm 0.00$ | $12.76 \pm 0.00$ | $12.76 \pm 0.00$ |
| Spartan | 100 | $13.40 \pm 0.00$ | $12.76 \pm 0.00$ | $12.76 \pm 0.00$ |
| | 300 | $13.43 \pm 0.00$ | $12.76 \pm 0.00$ | $12.76 \pm 0.00$ |

## F    FLOP-Sensitive Pruning

We demonstrate FLOP-sensitive pruning with Spartan on ResNet-50 using the following cost model: assign a cost of 1 to each parameter of a fully connected layer, and a cost of $N^2$ to each parameter of a convolutional layer where the output has size $N \times N$ along its spatial dimensions. We evaluate two valuation functions: $v_1(c_i, \theta_i) = c_i|\theta_i|$ and $v_{0.5}(c_i, \theta_i) = \sqrt{c_i}|\theta_i|$. $v_1$ results in the same pruning order as in standard pruning, but with a FLOP budget constraint instead of the usual sparsity budget. $v_{0.5}$ assigns a lower value to the parameters of convolutional layers, and results in networks where the parameters of convolutional layers are preferentially pruned. We use $\beta_{\max} = 10$ for $v_1$ and $\beta_{\max} = 160$ for $v_{0.5}$ to compensate for the relatively smaller scale of the normalized values $v/c$ in the soft top-$k$ forward pass (Algorithm 4).

Table 8 gives the top-1 accuracy, FLOP percentage, and sparsity percentage for each of these valuation functions. Spartan yields models with identical FLOP percentages of $5.76\%$, which is slightly higher than the budgeted value of $5\%$—this discrepancy is due to the additional cost of the normalization layers and activation functions in the network. Most notably, there is a substantial difference in the sparsity percentages realized by these valuation functions. As expected, $v_{0.5}$ preferentially sparsifies the parameters of convolutional layers and yields denser fully connected layers, resulting in lower sparsity overall.

Table 8: Results of FLOP-sensitive pruning experiments on ImageNet-1K with ResNet-50 models.

| | Accuracy % | FLOP % | Sparsity % |
|---|---|---|---|
| $v_1$ | 73.88 | 5.76 | 95.94 |
| $v_{0.5}$ | 74.13 | 5.76 | 89.73 |

[1]This setup for Top-KAST follows the protocol used by Jayakumar et al. [24]

# G   Additional Experiments

In Table 9, we compare Spartan against two additional variants of the Top-KAST baseline: Top-KAST with the Erdos-Renyi-Kernel (ERK) sparsity distribution, and with pruning applied to the parameters of all convolutional and fully-connected layers with the exception of bias terms (prune all). Top-KAST (excl. input conv.) denotes the Top-KAST variant used in the experiments presented in the main text, where we exclude the input convolutional layer from pruning. We find that there is some small variation in the measured top-1 validation accuracies, but our conclusion that Spartan improves generalization at higher levels of sparsity is unchanged.

Table 9: Comparison between Spartan and additional variants of the Top-KAST baseline on ImageNet-1K with ResNet-50 models.

|  |  | Sparsity | |
| --- | --- | --- | --- |
| Method | Epochs | 90% | 95% |
| Top-KAST (ERK) | 100 | 75.98 | 73.63 |
|  | 200 | 77.06 | 75.08 |
|  | 400 | 77.48 | 75.62 |
| Top-KAST (prune all) | 100 | 75.74 | 73.72 |
|  | 200 | 76.77 | 75.07 |
|  | 400 | 77.44 | 75.77 |
| Top-KAST (excl. input conv.) | 100 | 75.48 $\pm 0.15$ | 73.51 $\pm 0.16$ |
|  | 200 | 76.84 $\pm 0.11$ | 75.20 $\pm 0.11$ |
|  | 400 | 77.37 $\pm 0.07$ | 75.90 $\pm 0.04$ |
| Spartan | 100 | 76.17 $\pm 0.10$ | 74.68 $\pm 0.24$ |
|  | 200 | 77.06 $\pm 0.13$ | 75.92 $\pm 0.01$ |
|  | 400 | 77.40 $\pm 0.06$ | 76.48 $\pm 0.20$ |

# H   Learned Sparsity Patterns

We observe a qualitative difference in the distribution of per-layer sparsities between ViT-B/16 models trained with unstructured sparsity and those trained with block structured sparsity (Figure 5). In particular, the output projections of self-attention layers under block structured pruning are significantly more dense in the later blocks of the network relative to unstructured pruning. The reasons for this difference are not immediately clear to us, and we leave further investigation of this phenomenon to future work.

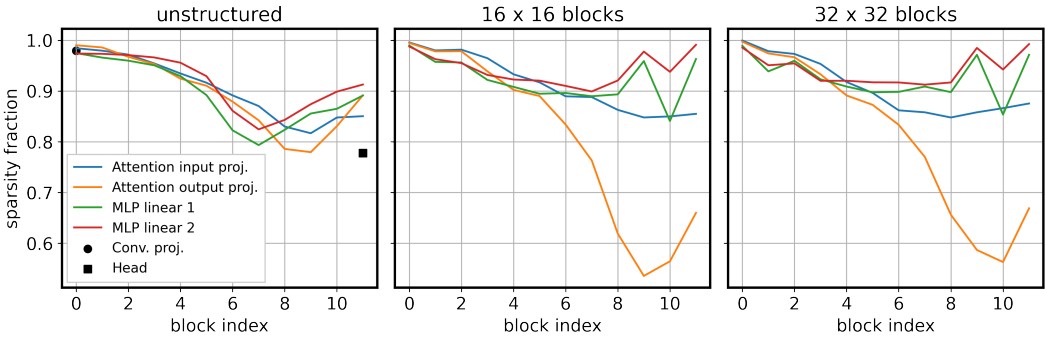

Figure 5: Per-layer sparsities of ViT-B/16 models trained with Spartan.

Block structured pruning produces coherent sparsity patterns in ViT-B/16 models. In Figure 6, we visualize the magnitudes of the weight matrices corresponding to the input projection of each

self-attention layer in a ViT-B/16 model trained with Spartan using $32 \times 32$ block structured pruning. This matrix maps vectors of dimension 768 to query, key, and value embedding vectors, each of dimension 768. We observe that the training process yields similar sparsity patterns in the query and key embedding submatrices, which correspond to the left and center panels in the visualization for each layer. This is an intuitively reasonable property since the self-attention layer computes inner products of the query and key embeddings in order to construct attention maps. We note that this symmetry emerges purely as a result of the optimization process; we did not incorporate any prior knowledge into Spartan regarding the role of particular entries of the weight matrices subject to sparsification.

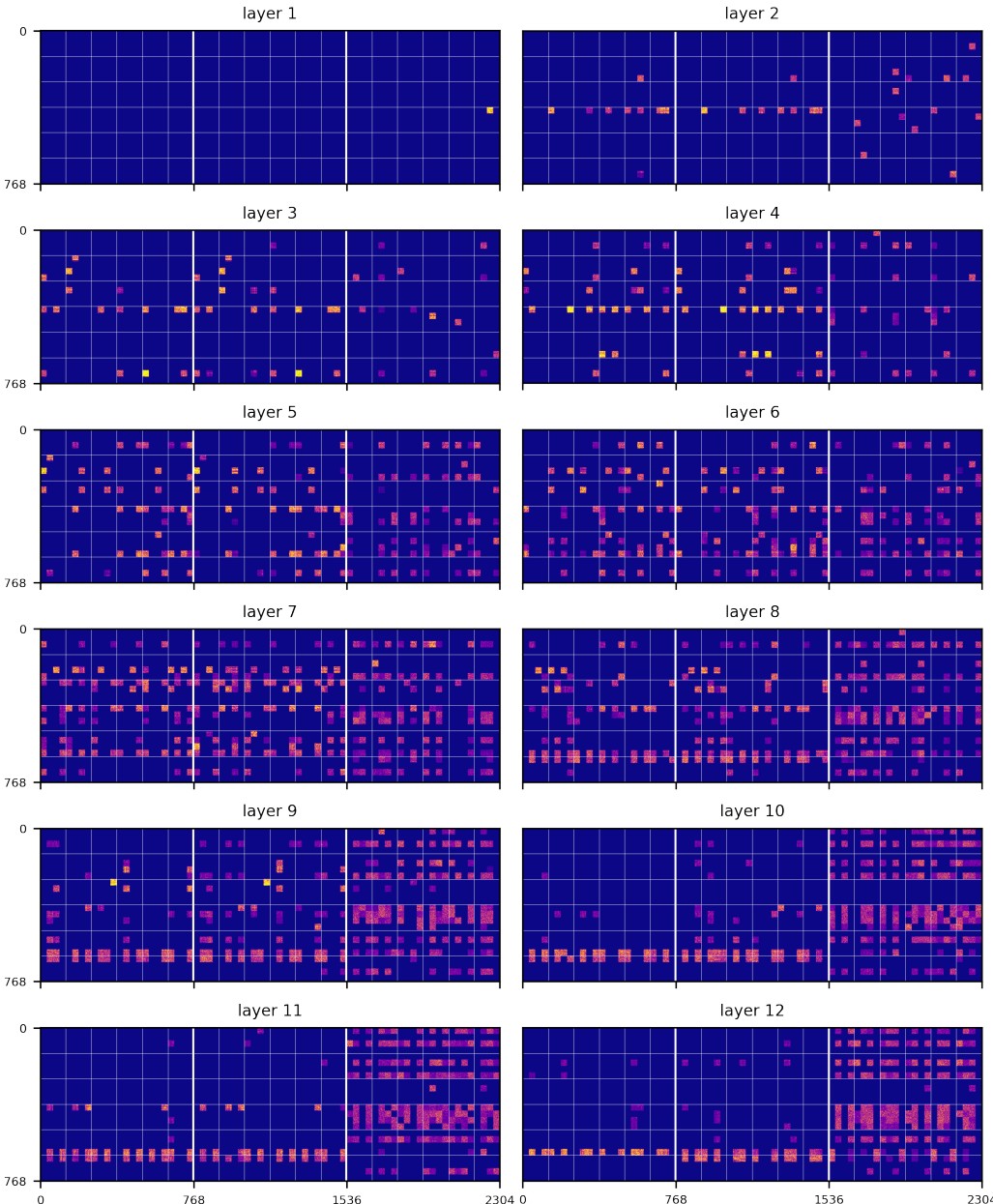

Figure 6: Weight magnitudes of the input projection matrices of each self-attention layer in a $32 \times 32$ block sparse ViT-B/16 network trained using Spartan.