# OpenReview forum: "Spartan: Differentiable Sparsity via Regularized Transportation"
_NeurIPS.cc/2022/Conference — NeurIPS 2022 Accept_

### Official Review · Reviewer_4MMR · 2022-06-24

**Rating:** 6
**Confidence:** 3
**Soundness:** 2 fair
**Presentation:** 3 good
**Contribution:** 2 fair

**Summary:**

The paper presents Spartan - a "dense-to-sparse" algorithm to train sparse neural networks in which the sparsity of parameters can be enforced directly by a predetermined budget.

The paper has $3$ main contributions:
- By incorporating the soft top-$k$ operator in the parameter update and by introducing the sharpness parameter, Spartan allows to interpolate between Iterative magnitude pruning and Top-$K$ Always sparse training.

- Spartan shows very competitive and consistent performance over the existing methods and also the fully dense training.

- The authors also study the effect of the sharpness parameter on the accuracy, as a trade-off between exploration and exploitation.

**Questions:**

- The problem $(2)$ is indeed an optimal transport problem, but it is not formulated in the standard form, i.e. what is the cost matrix, the admissible coupling (transport plan), the two marginal constraints? This makes it difficult to calculate and justify the update of Sinkhorn iterations.

- Please specify which kind of entropic regularization you are using. More precisely, $[4]$ use KL divergence but $[11]$ use the negative entropy. The two types are closely related, but mathematically, they result in a bit different Sinkhorn iterations. Based on the form of the mask $m$, I guess you are using the negative entropy, but not the KL divergence.

- In the Appendix, please consider to provide the details of the:
   + Soft top-$k$ operator in $[41]$. It is the core component of Spartan, so I believe it deserves some brief description.
   + Calculation of the backward pass. Even though the authors cite theorem $3$ in $[41]$, it is complicated for outsider, thus difficult to justify if the calculation in the Algorithm 5 is correct or not.
  + Calculation of the forward pass. Please also specify the entropic-regularization problem to which the authors apply.

- What is the dimension $d$ of the input vectors in the applications?

- How do the authors come up with the specific form of the initialization $-\beta |\theta_{i_k}| / c_{i_k}$?
Does it come from the empirical observation, or some heuristic?

- I see the definition of the projection $\Pi_k$ but wonder how is it implemented in practice? Is it simply keeping the largest $k$ elements and setting all others to zero?

**Ethics Review Area:**

["I don’t know"]

**Limitations:**

The authors do discuss the limitations and potential negative societal impact of their work.

**Strengths And Weaknesses:**

Amongst core components of Spartan, the algorithm 4 leaves me much doubt.

- It appears to me that the algorithm 4 is nothing but the Sinkhorn algorithm. In that case, it is not clear to me the motivation of the stopping criteria in the line 7. In practice (and also in the optimal transport community), it is enough to control the marginal violation (see Remark 4.6 in [CompOT]).

 - The calculation of the Sinkhorn iterations does not look right to me. Here, I present my own derivation of the Sinkhorn update. First, let us define $Y := [y, y'] \in \mathbb R^{d \times 2}$ by stacking two vectors in $\mathbb R^d$ and $C := [\frac{V}{c}, 0_d] \in \mathbb R^{d \times 2}$, where $0_d$ is the zero vector in $\mathbb R^d$. Then clearly, $\langle C, Y \rangle = (\frac{V}{c})^T y$, where $\langle C, Y \rangle = \sum_{ij} C_{ij} Y_{ij}$ denotes the Frobenius product. We can rewrite the problem $(2)$ as

$$
\min_{Y \in \mathbb R^{d \times 2}_+} \langle -C, Y \rangle \text{ subject to } Y 1_2 = c \text{ and } Y^T 1_d = (k, 1_d^T c - k)^T.
$$

The two marginals are not yet normalized. So, denote $s = 1_d^T c$, then define $c^* := \frac{c}{s}, C^* := sC = [\frac{V}{c^*}, 0_d]$ and $k^* := \frac{1}{s} (k, 1_d^T c - k)^T = ( \frac{k}{s}, 1 - \frac{k}{s})^T$. The above problem is equivalent to

$$
\min_{Y \in \mathbb R^{d \times 2}_+} \langle -C^*, Y \rangle \text{ subject to } Y 1_2 = c^* \text{ and } Y^T 1_d = k^*.
$$

Now, this is a properly defined optimal transport problem. If we define the entropic regularization problem as: for $\beta > 0$,

$$
\min_{Y \in \mathbb R^{d \times 2}_+} \langle -C^*, Y \rangle + \frac{1}{\beta} H(Y) \text{ subject to } Y 1_2 = c^* \text{ and } Y^T 1_d = k^*.
$$

where $H(Y) = \sum_{ij} Y_{ij} (\log Y_{ij} - 1)$. The Sinkhorn update now reads: for dual vectors $u \in \mathbb R^2$ and $v \in \mathbb R^d$, we have: $v = \log c^* - \log \sum_j \exp(\beta C^*_{\cdot, j} + u_j)$ and
$u = \log k^* - \log \sum_i \exp(\beta C^*_{i, \cdot} + v_i)$, and the optimal plan $Y$ is given by: $Y_{ij} = \exp(v_i + u_j + \beta C^*_{ij})$.

Comparing $v, u, Y$ to $\nu, \mu, m$ from the lines 4,5,6 in the Algorithm 4, respectively, I do not see why they are equivalent.

Reference:

[CompOT] Gabriel Peyré and Marco Cuturi (2019), "Computational Optimal Transport: With Applications to Data Science", Foundations and Trends in Machine Learning: Vol. 11: No. 5-6, pp 355-607.

---

> ### Author Response · Authors · 2022-08-02
> **Response to Reviewer Comments**
>
> We thank the reviewer for their considered comments on our submission.
>
> > It appears to me that the algorithm 4 is nothing but the Sinkhorn algorithm.
>
> Algorithm 4 is indeed an application of the Sinkhorn algorithm, as stated in L138: “we [...] solve the resulting regularized problem using the Sinkhorn-Knopp algorithm.”
>
> > It is not clear to me the motivation of the stopping criteria in the line 7. In practice (and also in the optimal transport community), it is enough to control the marginal violation (see Remark 4.6 in [CompOT]).
>
> As is typical for optimization methods in general, several different stopping criteria are used in practice. One such choice is the marginal violation, as indicated in the reviewer’s comments. Another choice is to stop when the relative change in the objective value is sufficiently small — for instance, as used in the experiments of Cuturi (2013) [11]. We use this latter option in Algorithm 4.
>
> > The calculation of the Sinkhorn iterations does not look right to me. [...] Comparing $v$, $u$, $Y$ to $\mu$, $\nu$, $m$ from the lines 4,5,6 in the Algorithm 4, respectively, I do not see why they are equivalent.
>
> The Sinkhorn updates in Algorithm 4 are correct as stated. To show this, we use the standard fact (see, e.g., Chapter 4.2 in [CompOT]) that we can always fix one dual variable to zero, due to there being one redundant constraint in the OT problem. Intuitively, this is equivalent to saying that we can always compute the $N$th row of the transportation plan given the remaining $N-1$ rows.
>
> Let us be explicit for the case of Problem 2 in the paper. Using the notation in the reviewer’s comments, we have that $Y_{i1} = \exp(\beta C^*_{i1} + v_i + u_1)$ and $Y_{i2} = \exp(v_i + u_2)$. Note that the dual variables in an optimal solution are not unique: for any dual optimal $(v^*, u^*)$ and $\delta \in \mathbb{R}$, $(v^* + \delta 1_d, u^* - \delta 1_2)$ is also dual optimal. Use this degree of freedom to fix $u_2 = 0$. This gives the updates $v = \log c^* - \log(1 + \exp(\beta sV/c + u_1))$  and $u_1 = \log k^* - \log \sum_i \exp(\beta sV_i/c_i + v_i)$. This matches the updates in Algorithm 4 up to a rescaling by the constant $s = 1^T c$, i.e., $c^* \rightarrow c$, $k^* \rightarrow k$ and $\beta s \rightarrow \beta$. Given that it is not necessary to normalize the marginals to sum to 1, we recover the updates in lines 4 and 5 in Algorithm 4 exactly. Finally, in line 6, we divide by the cost vector $c$ to recover the solution to Problem 1 from the solution to Problem 2, as required by the substitution defined in L133.
>
> We note as well that by fixing $u_2 = 0$, we effectively halve the computational cost of each Sinkhorn iteration by eliminating the need to compute $u_2 = \log(1^T c - k) - \log \sum_i \exp(v_i)$. We will add a section to the Appendix that clarifies this detail of our implementation.
>
> Additionally, we have added a section in the **rebuttal revision (Appendix G)** that details the derivation of the soft top-k backward pass (Algorithm 5).
>
> > Please specify which kind of entropic regularization you are using. More precisely, [4] use KL divergence but [11] use the negative entropy. The two types are closely related, but mathematically, they result in a bit different Sinkhorn iterations.
>
> We have revised Sec. 3.1 in the paper to explicitly state that we apply the standard entropic regularizer defined by $H(X) = -\sum_{ij} X_{ij} (\log X_{ij} - 1)$.
>
> Contrary to our understanding of the reviewer’s comments, we note that the entropic regularization perspective from [11] and the iterative Bregman projection perspective from [4] yield equivalent Sinkhorn updates (see, e.g., Section 3.1 in [4], or Remark 4.8 in [CompOT]). Since we might be missing some context here, we invite the reviewer to elaborate on the statement that these two perspectives yield different Sinkhorn updates.
>
> > What is the dimension $d$ of the input vectors in the applications?
>
> As stated in L181, the ResNet-50 and ViT-B/16 architectures consist of 25.6M and 86.4M parameters respectively. Thus, the dimension $d$ is respectively 25.6M and 86.4M for each of these networks in the unstructured case. In the block structured case, the dimension is divided by the number of elements in each block.
>
> > How do the authors come up with the specific form of the initialization $-\beta |\theta_{i_k}| / c_{i_k}$?
>
> This is a heuristic initialization based on the observation that the dual variable $\mu$ defines a threshold value analogous to the bias parameter in univariate logistic regression. Since we are aiming for an approximately k-sparse output, a natural guess for the value of this threshold is the top-$k$th entry of $z$, i.e., $\beta |\theta_{i_k}| / c_{i_k}$. We will revise the manuscript to clarify this reasoning.
>
> > I see the definition of the projection $\prod_k$ but wonder how is it implemented in practice?
>
> $\prod_k$ is implemented by zeroing out the smallest $d-k$ elements by magnitude.

---

> > ### Comment · Reviewer_4MMR · 2022-08-07
> > **Response to Authors**
> >
> > I thank the authors for their reponse.
> >
> > > Contrary to our understanding of the reviewer’s comments, we note that the entropic regularization perspective from [11] and the iterative Bregman projection perspective from [4] yield equivalent Sinkhorn updates (see, e.g., Section 3.1 in [4], or Remark 4.8 in [CompOT]). Since we might be missing some context here, we invite the reviewer to elaborate on the statement that these two perspectives yield different Sinkhorn updates.
> >
> > Indeed, I did not cite the correct reference and thank the authors for pointing out. I intended to say that the KL divergence also can be used as regularizer (see Remark 4.2 in [CompOT]). In that case, it will result in a bit different Sinkhorn updates than those in [4],[11]. I also agree that [4] and [11] yield the same update.
> >
> > > This gives the updates $v = \log c^* - \log \Big[ 1 + \exp \big( \frac{\beta s V}{c} + u_1 \big) \Big]$ and $u_1 = \log k^* - \log \sum_i \exp \Big[ \frac{\beta s V_i}{c_i} + v_i \Big]$. This matches the updates in Algorithm 4 up to a rescaling by the constant $s = 1^T c$, i.e. $c^* \to c$, $k^* \to k$,  and $\beta s \to \beta$. Given that it is not necessary to normalize the marginals to sum to 1, we recover the updates in lines 4 and 5 in Algorithm 4 exactly. Finally, in line 6, we divide by the cost vector  to recover the solution to Problem 1 from the solution to Problem 2, as required by the substitution defined in L133.
> >
> > I agree with the authors response up to the previous part of this paragraph. I don't understand the rescaling step. Taking, for example $u_1$: The update in line 5 in Algo 4 reads $u_1 = \log k - \log \sum_i \Big[  \exp \big( \frac{\beta V_i}{c_i} + v_i \big)  \Big]$. On the other hand, from the authors response,
> > we can rewrite the update as $u_1 = \log k - \log s - \log \sum_i \exp \Big[ \frac{\beta s V_i}{c_i} + v_i \Big] = \log k - \log  \Big[ s \sum_i \exp \big( \frac{\beta s V_i}{c_i} + v_i \big) \Big]$. Maybe I'm missing something here but I am still not clear how the variable $s$ can disappear after the rescaling.
> >
> > Moreover, regarding the calculation of the first column of optimal plan, $Y_{i1} = \exp \Big( \frac{\beta s V_i}{c_i} + u_1 + v_i \Big) $, but the line 6 in Algo 4 reads $Y_{i1} = \exp \Big( \frac{\beta V_i}{c_i} + u_1 + v_i \Big) $ (assuming that we always talk about the problem $2$, so no need to be divided by the cost $c$). How can the variable $s$ can be eliminated by rescaling?
> >
> > It is possible that we are not talking about the same variables, i.e. my $v$ is not the same as $\nu$, my $u_1$ is not the same as $\mu$ and my $Y_{\cdot,1}$ is not the same as $m$ (up to a multiplicative constant). In that case, probably it would be better if the authors can write down their own derivation, so that it is easier to discuss and compare.

---

> > > ### Author Response · Authors · 2022-08-08
> > > **Derivation of Sinkhorn updates**
> > >
> > > On scaling: all we are saying here is that it is not necessary to normalize the marginals to sum to 1 (as long as the marginals have strictly positive entries, the entropy is well-defined). Therefore, we can simply set $s=1$ to obtain the stated Sinkhorn updates.
> > >
> > > Here is a derivation of the update using the notation from the paper:
> > >
> > > The cost matrix is $C = \left[ -\frac{v}{c} , 0 \right] \in \mathbb{R}^{d\times 2}$, the row marginals are $c$ and the column marginals are $[k, 1_d^Tc - k ]$. From [11, Lemma 2], we know that the optimal solution to Problem 2 can be written in the form $\mathrm{diag}(\exp\nu) \exp(-\beta C) \mathrm{diag}(\exp [\mu, \mu'])$, with dual variables $\mu, \mu' \in \mathbb{R}$ and $\nu \in \mathbb{R}^d$. Moreover, we can compute a sequence of iterates converging to an optimal collection of dual variables using Sinkhorn iteration.
> > >
> > > Use the degree of freedom in the dual variables to fix $\mu' = 0$.
> > >
> > > The Sinkhorn updates are therefore:
> > >
> > > $\nu_{t+1} = \log c - \log (\exp(-\beta C_{\cdot 1} + \mu_t) + \exp(-\beta C_{\cdot 2} + \mu'_t)) = \log c - \log(1_d + \exp(\beta v / c + \mu_t)),$
> > >
> > > $\mu_{t+1} = \log k - \log \sum_i \exp(-\beta C_{i1} + \nu_{t,i}) = \log k - \log \sum_i \exp(\beta v_i / c_i + \nu_{t,i}).$

---

> > > > ### Comment · Reviewer_4MMR · 2022-08-09
> > > > **Response to Authors**
> > > >
> > > > I thank the authors for their response.
> > > >
> > > > It is all clear to me now. The authors have addressed all my concerns. I believe that these clarification and details should be included in the final version, if the paper is accepted.
> > > >
> > > > I am happy to raise the score to 6.

---

> > > > > ### Author Response · Authors · 2022-08-09
> > > > > **Thanks**
> > > > >
> > > > > We greatly appreciate this and would like to gently remind the reviewer to actually raise the rating to 6 as it still seems to be set at 3.

---

### Official Review · Reviewer_X3Jw · 2022-07-05

**Rating:** 6
**Confidence:** 3
**Soundness:** 3 good
**Presentation:** 3 good
**Contribution:** 3 good

**Summary:**

This paper proposes a new method for pruning DNN parameters. The proposed method uses a differential top-K operator so that the proposed method interpolates between IMP and Top-KAST and balances exploitation and exploration well.

**Questions:**

How was $-\beta |\theta_{i_k}| / c_{i_k}$ (i.e., the initial value) derived?

**Limitations:**

The authors mentioned the limitations and social impacts well in Section 5.

**Strengths And Weaknesses:**

# Strengths

- The proposed method is simple and easy to use.
- The proposed method outperforms existing methods.
- The experiments include the modern ViT architecture.

# Weaknesses

- While I grasp the intuitive idea of the proposed method, e.g., the proposed method interpolates IMP and Top-KAST, I don't find the rationale behind the specific design of the proposed update rule. More detailed discussions and hopefully principled interpretation, e.g., the update of the proposed method can be interpreted as a descent step of some objective function, should be given.

---

> ### Author Response · Authors · 2022-08-02
> **Response to Reviewer Comments**
>
> We appreciate the reviewer’s feedback on our submission.
>
> > While I grasp the intuitive idea of the proposed method, e.g., the proposed method interpolates IMP and Top-KAST, I don't find the rationale behind the specific design of the proposed update rule. More detailed discussions and hopefully principled interpretation, e.g., the update of the proposed method can be interpreted as a descent step of some objective function, should be given.
>
> Algorithms 1-3 aim to optimize the following constrained problem: “minimize $L(\theta)$ subject to $\lVert\theta\rVert_0 = k$,” where $L$ is the empirical loss. Spartan can be interpreted as a combination of (1) the dual averaging method, as previously applied in Top-KAST, and (2) a homotopy method via the use of the soft top-k function.
>
> The use of dual averaging is principled insofar as its use is well-justified for constrained convex optimization (see, e.g., [40] and the references therein). On the other hand, homotopy methods in non-convex optimization are heuristics that aim to “guide” the optimization trajectory towards better local minima via progressive relaxations of the original problem (see, e.g., [SKC06] for an ML application of a homotopy method). In the case of Spartan, the heuristic use of the soft top-k operator (in tandem with the scheduling of the $\beta$ hyperparameter) is similar in spirit to the longstanding technique in optimization of annealing a temperature term over time.
>
> We will add some additional discussion of this perspective on the method to the manuscript.
>
> [SKC06] Sindhwani, Keerthi, Chapelle. Deterministic Annealing for Semi-supervised Kernel Machines. ICML 2006.
>
> > How was $-\beta |\theta_{ik}| / c_{ik}$ (i.e., the initial value) derived?
>
> This is a heuristic initialization based on the observation that the dual variable $\mu$ defines a threshold value analogous to the bias parameter in univariate logistic regression. Since we are aiming for an approximately k-sparse output, a natural guess for the value of this threshold is the top-$k$th entry of $z$, i.e., $\beta |\theta_{i_k}| / c_{i_k}$. We will revise the manuscript to clarify this reasoning.

---

> > ### Comment · Reviewer_X3Jw · 2022-08-09
> > **Thank you**
> >
> > Thank you for the response. I keep my score and vote for weak acceptance.

---

### Official Review · Reviewer_1csR · 2022-07-05

**Rating:** 8
**Confidence:** 3
**Soundness:** 3 good
**Presentation:** 3 good
**Contribution:** 3 good

**Summary:**

This article introduces a new method, Spartan, which allows training neural networks with sparsity constraints. Spartan belongs to the "dense-to-sparse" family of methods: it maintains a dense parameter during training and makes it sparse little by little. Spartan controls the level of sparsity thanks to a parameter $\beta$ which performs an interpolation between two popular methods to train sparse neural networks (Top-KAST ref [23] of the paper at $\beta = 0$ and IMP [47] at $\beta = +\infty$). The central idea is to use, during training, a "mask" on the network parameters based on a soft-topk operator (implemented via regularized optimal transport).


**Questions:**

1° Relations with Top-KAST:

It is said in the introduction that the Top-KAST method [47] is a "dense-to-sparse" method, i.e. it maintains in memory a more or less dense parameter during the iterations. This point seems to me quite surprising because in [47] the authors specify that the ultimate goal of Top-KAST is to maintain sparsity during training ("In this work we propose Top-KAST, a method that preserves constant sparsity throughout training (in both the forward and backward-passes)"). Can the authors clarify this point?


2° About notations:

There is an important notation which comes up regularly but which is not explained and difficult to catch. In Algorithm 1 and Algorithm 3 at the backward pass the gradient with respect to a vector is used and multiplied with the gradient of the loss.

For example in Algorithm 1 line 2 $\theta_{t+1} = \theta_t - \eta_t \nabla \tilde{\theta}_t \nabla L(\tilde{\theta}_t)$. I don't think this notation is really standard. If I read between the lines, I understand this notation as $\nabla \tilde{\theta}_t=$ "support of $\tilde{\theta}_t$" (i.e. $0$ or $1$) and the multiplication with $\nabla L(\tilde{\theta}_t)$ as being a point-to-point multiplication by the support mask. More precisely $\nabla \tilde{\theta}_t \nabla L(\tilde{\theta}_t) = \nabla L(\tilde{\theta}_t)$ if $\tilde{\theta}_t \neq 0$ and $0$ otherwise.

Is this correct? I think this point deserves more details because it is difficult to understand  what is done in practice.

3° About Algorithm 3:

I have trouble understanding the interest of line 2 in algorithm 3: after finding $\sigma^{\beta}_k(\theta_t)$ by a soft-topk operator on $\theta_t$ (and as shown in figure 2) an additional step of projection onto $k$-sparse set is added $\tilde{\theta_t} = \Pi_k(\sigma^{\beta}_k(\theta_t))$ and the gradient of the loss is computed with respect to this $\tilde{\theta_t}$. It seems to me that the two are quite redundant.

It is argued in the article that this is to "mitigate the issue of gradient sparsity" and an ablation study is conducted on this point in section 4.2 to justify this step. I find these arguments quite empirical and it is difficult to feel the underlying intuition. Do the authors have some theoretical points to support them? For example, can we show for a simple function that the optimization "goes wrong" if we do not add this top-k step?

Also, it seems to me that if this step was not there, then when $\beta = 0$ Spartan would no longer be equivalent to Top-KAST: am I correct ?

4° Experimental section:

The experiments are in my opinion convincing. The idea of alternating an exploration and exploitation phase is sounded and very well illustrated by Figure 3. The experiments show well the interest of Spartan from the point of view of performance vs sparsity. Although Spartan does not outperform Top-KAST much on ResNet-50 (of the order of the variance in most cases) the ViT experiment shows the opposite and Spartan has very good performance especially in the structured case.

The computational overhead compared to dense training seems reasonable to me. On this point I think it is also important to put the Top-KAST one in perspective on the same figure: how do they compare?


5° Small remarks:
- The "sparsity mask" is never clearly defined. We understand that it is $m \in [0,1]^{d}$ but I think it is important to say it clearly.
- Figure 1 (c) doesn't look good in pdf because I think it's a .png or .jpg : it could be interesting to recompile the figure in pdf so that it's more readable.
- I think that the link between equation (2) and the optimal transport could be more detailed (e.g. in appendix). Having the formulation "min on a coupling" could help understanding for readers who are not familiar with optimal transport.
- What are "crude linear cost models" (Section 5: discussion) ?

To conclude I would like to note that it is quite rare to have a "Limitations" paragraph and I find this a good approach that I welcome (especially since the remarks made by the authors are good remarks that also help in understanding the method).


------ AFTER REBUTTAL ------

I thank the authors for their response which addressed most of my concerns, therefore I change my score to 8.

**Strengths And Weaknesses:**

Overall I find this article very well written, especially the introduction, the background and the presentation of the contributions. The authors have made a notable effort of pedagogy which greatly facilitates the reading and the understanding of the article. In particular, figures 1 and 2 are very clear, didactic and I thank the authors for this effort. On the other hand, the proposed method seems to me quite relevant, and well supported by many experiments. The fact that Spartan is linked to and generalizes IMP and Top-KAST strengthens the contribution. The method seems to me quite new: it skillfully combines the soft-topk operator of Xie et al [41] with dual averaging ideas.

However, there are a few points that are, in my opinion, unclear and would need to be clarified. These prevent me from putting a very clear accept on this article, but I would gladly change my mind depending on the authors' answers.

---

> ### Author Response · Authors · 2022-08-02
> **Response to Reviewer Comments**
>
> We thank the reviewer for their consideration of our submission.
>
> > 1° Relations with Top-KAST
>
> Top-KAST maintains a dense parameter vector in memory during training, and uses a sparse, top-k masked version of these parameters to compute the forward and backward passes (see the last paragraph of Section 2.1 in [47] — “$\alpha^t$ is best thought of as a ``temporary view’’ of the dense parameterisation, $\theta^t$” — and Algorithm 1 in Appendix D). Thus, it incurs dense memory cost but sparse computational cost. We will revise our use of the “dense-to-sparse”/”sparse-to-sparse” nomenclature in the introduction, since this terminology is somewhat ambiguous (it would be more precise to separate the computational and memory costs of these algorithms).
>
> > 2° About notations
>
> Correct, $\nabla \tilde{\theta}_t \nabla L(\tilde{\theta}_t)$ should be interpreted as “compute the gradient of the loss with respect to the sparse parameters $\tilde{\theta}_t$, and then apply the same k-sparse mask used to obtain $\tilde{\theta}_t$ to the gradient”. Here, $\nabla \tilde{\theta}_t$ denotes the Jacobian of $\Pi_k (\theta_t)$ w.r.t. $\theta_t$, which is simply a $d \times d$ matrix with the entries of the mask on the diagonal. We will clarify the notation here to avoid any potential confusion.
>
> > 3° About Algorithm 3
>
> > I have trouble understanding the interest of line 2 in algorithm 3 [...] It seems to me that the two are quite redundant.
>
> Without the additional k-sparse projection, the parameters are only approximately sparse after the application of the soft top-k operator. We find that the “leakage” of information through these non-zero parameters can result in a significant train-test mismatch, since at test time we always use an exactly k-sparse network. The addition of the k-sparse projection in the forward pass during training helps to mitigate this gap. This effect can be seen in Figure 4 (left) as the gap between the blue and orange lines at smaller values of $\beta$.
>
> >  I find these arguments quite empirical and it is difficult to feel the underlying intuition. Do the authors have some theoretical points to support them? For example, can we show for a simple function that the optimization "goes wrong" if we do not add this top-k step?
>
> Unfortunately, we do not currently have a theoretical justification as to why Spartan achieves better generalization error than iterative magnitude pruning and Top-KAST. More broadly, this is a difficult and fundamental problem in the field: there is very limited theoretical understanding supporting the use of particular optimizers for deep networks even in the standard supervised setting, let alone with the additional complication of sparsity constraints.
>
> > Also, it seems to me that if this step was not there, then when $\beta = 0$ Spartan would no longer be equivalent to Top-KAST: am I correct ?
>
> This is correct.
>
> > 5° Small remarks:
> > What are "crude linear cost models" (Section 5: discussion) ?
>
> This just refers to our use of linear cost models of the form $\mathrm{cost} = c^T m$ for a cost vector $c$ and sparsity mask $m$. This is “crude” in the sense that such a cost model doesn’t include interaction terms that encode, e.g., the notion that coherently pruning an entire row or column of a parameter matrix will yield more cost savings than pruning random entries of the matrix.

---

> > ### Comment · Reviewer_1csR · 2022-08-03
> > **Response to authors**
> >
> > I thank the authors for their response. I believe they answer all the concerns that I had, especially regarding the notations and the the “dense-to-sparse”/”sparse-to-sparse” nomenclature. l think both should be clarified in the final version if the manuscript is accepted.
> >
> > Overall I think this is a serious work with good contributions and solid background. The fact that authors wrote the OT problem in "standard form" at the end make the article a bit more self-content with respect to [41] and that is a good thing.
> >
> > Therefore I increase my score to 8.

---

### Official Review · Reviewer_HMyH · 2022-07-11

**Rating:** 5
**Confidence:** 3
**Soundness:** 3 good
**Presentation:** 4 excellent
**Contribution:** 3 good

**Summary:**

This paper proposes a gradual neural network pruning algorithm which utilizes soft top-k mechanism for the weight update of masked/unmasked parameters. In particular, authors adopt the differentiable top-k mechanism of Xie et al. (2020), which smooths the top-k problem in a principled manner via viewing top-k as an optimal transport problem and adds the entropic regularizer to make the allocation smooth. The model applies both soft and hard top-k for the evaluation of the model, and only soft for the gradient computation. When combined with an appropriate scheduling of sparsity and entropic regularization intensity, the method achieves SOTA-like performance on ResNet-50 and ViT-B trained on ImageNet-1K.

**Questions:**

- Up to my knowledge, Top-KAST has two different types of sparsity; forward sparsity and backward sparsity. I wonder what backward sparsity the authors used for reproducing the top-KAST results. Are you using fully dense backwards?

- I am curious where the inference FLOP benefits of Spartan comes from. Could authors provide a layerwise sparsity plot? Also, how would the inference FLOPs of top-KAST be if one uses ERK layerwise sparsities?

- I believe that the inference FLOP gain is a very strong benefit of the proposed method. I suggest the authors to move them to the main text if possible.

**Limitations:**

Authors adequately addressed these, in my humble opinion.

**Strengths And Weaknesses:**

__Strength.__ First of all, I must say that the paper is very clearly written; I like how algorithms 1,2 are placed, which makes algorithm 3 very sensible and straightforward thing to do. Also, the "dual averaging"-based perspective toward the top-KAST update was quite new and fresh to me (although I would have appreciated slightly more detailed discussion on what dual averaging is), and I think this could be inspirational for many future works. Another big strength of the proposed method is its inference efficiency, having much lower (theoretical) inference FLOP than top-KAST.

__Weakness.__ My main concern is on the _practicality_ of the proposed method. Although the central idea of the algorithm is interesting and it seems like it gives slight boost in terms of the final sparsity-to-performance tradeoff, I am slightly worried about the practicality of the method. Spartan has additional hyperparameter $\beta$ which needs to be carefully selected and scheduled for the performance gain over the existing methods (which may require many training runs to be tuned). Also, unlike magnitude pruning (whose active parameters decrease as the training proceeds) and top-KAST (which has decreased number of peak #parameters via backward sparsity), Spartan has all parameters active throughout the training, having very little benefit in terms of resource used for training (training FLOPs, memory).

Here are some nitpicks (or much minor-er concerns):
- In line 60, the paper mentions that "We show that Spartan interpolates between ..." but I do not think this point has been rigorously shown. In fact, a key idea of top-KAST is using backward sparsity, which is slightly less than forward sparsity but not dense, and Spartan uses the fully dense(-but-rescaled) backward.
- In line 137, authors seem to argue that the entropic regularization is necessary in order to efficiently solve Eq.(2) (or equivalently Eq. (1)). I do not think this is necessarily true. In fact, the solution of Eq.(1) is actually quite easy to get; for uniform c, one would simply need to perform the magnitude-based pruning (or cost-rescaled magnitude whenever c is non-uniform). Thus, I view the entropic regularization as an artificially introduced (but nevertheless neat) artifact to make the allocation softer, instead of a requirement.

---

> ### Author Response · Authors · 2022-08-02
> **Response to Reviewer Comments**
>
> We appreciate the feedback on our submission. Our responses to the specific points raised in the review are as follows.
>
> > I am slightly worried about the practicality of the method [...] Spartan has additional hyperparameter $\beta$ which needs to be carefully selected and scheduled for the performance gain over the existing methods
>
> While $\beta$ does have to be tuned to optimize the performance of Spartan, we empirically observe that Spartan improves on our baselines over a wide range of $\beta$ values. This can be seen in Figure 4 (left), where Spartan (orange line) improves on Top-KAST (dashed line) over an order of magnitude of $\beta$ values: from $\beta \approx 1$ to $\beta \approx 40$. This suggests that $\beta$ does not have to be precisely fine-tuned in order to realize gains over existing methods. From our experiments with ResNet and ViT architectures, we recommend a default range of $\beta_\mathrm{max} \in [10, 20]$ for unstructured sparse training, and the same range scaled by the block size for block structured sparse training (as explained in L206-209).
>
> We additionally note that Spartan yields a substantial improvement in accuracy over Top-KAST in training block sparse ViT networks (Table 3). Given the increasing adoption of ViT-style architectures and the practical importance of block sparsity as a GPU-friendly form of parameter sparsity, we believe that Spartan is a useful tool for practitioners looking to mitigate the high inference cost of transformer-based architectures.
>
> > Spartan has all parameters active throughout the training, having very little benefit in terms of resource used for training (training FLOPs, memory).
>
> Spartan indeed incurs a higher computational cost during training compared to iterative magnitude pruning and Top-KAST due to its use of the soft top-$k$ operator (for our implementation, we measure a 5% overhead in per-iteration wall clock time over dense training, as described in Section 4.2).
>
> While training efficiency is not a focus of our work, we remark that Spartan is amenable to additional training-time optimizations that will improve its computational efficiency. As in Top-KAST, Spartan can take advantage of sparse kernels in the forward pass due to the top-k sparsification of the parameter vector (line 2 in Algorithm 3). In the backward pass, Spartan can similarly make use of sparse kernels when backpropagating gradients with respect to the activations of each layer (however, computing gradients with respect to the parameters will still incur a dense computational cost).
>
> As for memory usage, Spartan incurs a similar cost to Top-KAST since both methods maintain a dense version of the parameter vector during training.
>
> > In line 60, the paper mentions that "We show that Spartan interpolates between ..." but I do not think this point has been rigorously shown. [...]  In fact, a key idea of top-KAST is using backward sparsity [...]
>
> To be more precise, Spartan interpolates between Top-KAST with zero backward sparsity and iterative magnitude pruning. This is due to the fact that the soft top-k operator varies from a constant scaling function at $\beta = 0$ to the hard top-k function at $\beta = \infty$. We specifically consider Top-KAST with zero backward sparsity since it yields the highest test accuracy (Figure 2(b) in [23]).
>
> > Thus, I view the entropic regularization as an artificially introduced (but nevertheless neat) artifact to make the allocation softer, instead of a requirement.
>
> This is correct — the implication in L137 that entropic regularization is necessary to efficiently solve Problem 2 is an inaccuracy on our part, and we have revised this sentence accordingly.
>
> > I wonder what backward sparsity the authors used for reproducing the top-KAST results. Are you using fully dense backwards?
>
> We use Top-KAST with a dense backward pass, since this yields the highest accuracy for the method (see Figure 2(b) in [23]).
>
> > I am curious where the inference FLOP benefits of Spartan comes from. Could authors provide a layerwise sparsity plot? Also, how would the inference FLOPs of top-KAST be if one uses ERK layerwise sparsities?
>
> The FLOP difference between Spartan and Top-KAST in Table 6 is due to Spartan allocating relatively higher sparsity to convolutional layers in earlier layers of the network whose outputs have larger spatial dimensions, and therefore larger per-parameter computational cost. The percentage inference FLOP cost of Top-KAST with ERK layerwise sparsities would be identical to those reported in [15, Figure 2] — i.e., 42% at 80% sparsity and 24% at 90% sparsity, which are higher than our measured FLOP costs for Top-KAST and ERK.

---

> > ### Comment · Reviewer_HMyH · 2022-08-07
> > **Thank you**
> >
> > Thank you for the detailed response, especially on the choice of $\beta$. I have read other reviews and responses, and will keep my current score (being slightly inclined for acceptance).

---

### Author Response · Authors · 2022-08-06
**Follow-up on Author Response**

Dear Reviewers,

Thank you once again for taking the time to provide feedback on our submission. We hope that our responses adequately addressed the concerns raised in your reviews. If not, we would be happy to further engage with you during the remainder of the current discussion period.

Kind regards,

The Authors

---

### Meta-Review · Area_Chair_1P8B · 2022-08-26

**Recommendation:** Accept
**Confidence:** Certain

**Metareview:**

All reviewers agree that the paper is clearly written and proposes an algorithm which is both novel and efficient.

The rebuttal has clarified a number of points, and thereby adressed most of the concerns of the reviewers. The authors are thus strongly encouraged to take into account the comments of the reviewers and to add some of the clarifications that they provided in this discussion in the paper and supplementary materials.


**Award:**

No

---

### Decision · Program_Chairs · 2022-09-14

Accept